# RalB directly triggers invasion downstream Ras by mobilizing the Wave complex

Giulia Zago[1,2], Irina Veith[1,2], Manish Kumar Singh[1,2], Laetitia Fuhrmann[1,3], Simon De Beco[1,4], Amanda Remorino[1,4], Saori Takaoka[1,2], Marjorie Palmeri[1,2], Frédérique Berger[1,5], Nathalie Brandon[1,2], Ahmed El Marjou[1,6], Anne Vincent-Salomon[1,3], Jacques Camonis[1,2], Mathieu Coppey[1,4], Maria Carla Parrini[1,2]*

[1]Institut Curie, Centre de Recherche, Paris Sciences et Lettres Research University, Paris, France; [2]ART Group, Inserm U830, Paris, France; [3]Department of Pathology, Institut Curie, Paris, France; [4]LOCCO Group, UMR168, Paris, France; [5]Department of Biostatistics, Institut Curie, Paris, France; [6]Protein Expression and Purification Core Facility, Paris, France

**Abstract** The two Ral GTPases, RalA and RalB, have crucial roles downstream Ras oncoproteins in human cancers; in particular, RalB is involved in invasion and metastasis. However, therapies targeting Ral signalling are not available yet. By a novel optogenetic approach, we found that light-controlled activation of Ral at plasma-membrane promotes the recruitment of the Wave Regulatory Complex (WRC) via its effector exocyst, with consequent induction of protrusions and invasion. We show that active Ras signals to RalB via two RalGEFs (Guanine nucleotide Exchange Factors), RGL1 and RGL2, to foster invasiveness; RalB contribution appears to be more important than that of MAPK and PI3K pathways. Moreover, on the clinical side, we uncovered a potential role of RalB in human breast cancers by determining that RalB expression at protein level increases in a manner consistent with progression toward metastasis. This work highlights the Ras-RGL1/2-RalB-exocyst-WRC axis as appealing target for novel anticancer strategies.
DOI: https://doi.org/10.7554/eLife.40474.001

*For correspondence:
maria-carla.parrini@curie.fr

**Competing interests:** The authors declare that no competing interests exist.

## Introduction

One of the most frequent oncogenic events in human cancers is the activation by constitutive mutations of Ras GTPases: K-Ras, H-Ras, N-Ras. Roughly 30% of all human tumors carry Ras genetic alterations; the Ras mutation frequency is particularly high for pancreas (91%), lung (33%) and colon (51%) cancers (cbioportal) (*Gao et al., 2013*; *Simanshu et al., 2017*). Up-to-date no effective targeted therapies can be offered to patients with tumors carrying Ras mutations. Since Ras proteins are still considered undruggable (*Cox et al., 2014*), the focus for anti-Ras drug discovery moved downstream. Ras activates three major downstream pathways: the MAP kinases, the PI3 kinases and the Ral GTPases. Despite the considerable translational research efforts on the downstream kinases, the results are very deceiving. For example, MAPK pathway inhibition with MEK inhibitor therapy turned out to be largely ineffective (*Jänne et al., 2013*).

On the other hand, the targeting of Ral GTPases has been a much less exploited strategy (*Yan et al., 2014*; *Yan and Theodorescu, 2018*). The two human Ral proteins (RalA and RalB) are activated by six RalGEFs (Guanine Nucleotide Exchange Factors) (*Neel et al., 2011*). Among the six, four RalGEFs (RalGDS, Rgl1, Rgl2, Rgl3) have a Ras-association (RA) domain and are direct effectors of Ras. Activated GTP-bound Ras recruits these RalGEFs at the plasma-membrane triggering the

**eLife digest** Cancers develop when cells in the body divide rapidly in an uncontroled manner. It is generally possible to cure cancers that remain contained within a small area. However, if the tumor cells start to move, the cancer may spread in the body and become life threatening. Currently, most of the anti-cancer treatments act to reduce the multiplication of these cells, but not their ability to migrate.

A signal protein called Ras stimulates human cells to grow and move around. In healthy cells, the activity of Ras is tightly controled to ensure cells only divide and migrate at particular times, but in roughly 30% of all human cancers, Ras is abnormally active. Ras switches on another protein, named RalB, which is also involved in inappropriate cell migration. Yet, it is not clear how RalB is capable to help Ras trigger the migration of cells.

Zago et al. used an approach called optogenetics to specifically activate the RalB protein in human cells using a laser that produces blue light. When activated, the light-controlled RalB started abnormal cell migration; this was used to dissect which molecules and mechanisms were involved in the process.

Taken together, the experiments showed that, first, Ras 'turns on' RalB by changing the location of two proteins that control RalB. Then, the activated RalB regulates the exocyst, a group of proteins that travel within the cell. In turn, the exocyst recruits another group of proteins, named the Wave complex, which is part of the molecular motor required for cells to migrate.

Zago et al. also found that, in patients, the RalB protein was present at abnormally high levels in samples of breast cancer cells that had migrated to another part of the body. Overall, these findings indicate that the role of RalB protein in human cancers is larger than previously thought, and they highlight a new pathway that could be a target for new anti-cancer drugs.

DOI: https://doi.org/10.7554/eLife.40474.002

activation of Ral by GDP to GTP exchange. The two remaining RALGEFs (RalGPS1 and RalGPS2) do not bind Ras; their specific activators are still unknown, but they do contain a pleckstrin homology (PH) domain (*Rebhun et al., 2000*), responsible for membrane targeting, and a proline-rich motif which binds to SH3-containing signaling proteins (*Ceriani et al., 2007*).

Even though, in some cellular contexts, RalA and RalB seem to have overlapping effects, several evidences pointed out a distinct role for RalB in supporting the invasiveness of transformed cells. In the case of migration of the UMUC3 human bladder cancer cell line (K-Ras mutated), RalA seems to even antagonize the pro-migratory function of RalB (*Oxford et al., 2005*). The requirement of RalB for invasion in vitro (by Transwell Invasion assay) was shown by shRNA knock-down approach in seven out of nine K-Ras mutated human pancreatic cancer cell lines (*Lim et al., 2006*). Consistently, another study showed that RalB, but not RalA, plays a role in invadopodia formation in human pancreatic cancer cell lines (*Neel et al., 2012*). RalB, but not RalA, is required for the contractility-driven invasion of lung cancer cells (A549, K-Ras mutated) (*Biondini et al., 2015*). Moreover, in vivo metastasis assays in mice (tail vein injection) (*Ward et al., 2001*; *Lim et al., 2006*) and in hamsters (subcutaneous injection) (*Rybko et al., 2011*) supported a function of RalB pathway in the formation of tumor metastasis, both in Ras-mutated and Rous sarcoma virus-transformed cells. Besides the cancer context, we previously found that RalB, but not RalA, controls the mesenchymal migration of normal cells (NRK, HEK-HT, wild-type for Ras) by mobilizing its effector exocyst (*Rossé et al., 2006*; *Parrini et al., 2011*; *Biondini et al., 2016*), which is an octameric protein complex involved in the tethering of secretory vesicles to the plasma membrane prior to SNARE-mediated fusion (*Wu and Guo, 2015*).

All these works allowed to recognize the relevance of RalB pathway for motility, invasion and metastasis, but the underlying molecular mechanisms remain elusive (*Neel et al., 2012*). We reasoned that the emerging optogenetics technology (*Toettcher et al., 2013*; *Tischer and Weiner, 2014*) might help to overcome this limitation, because of its capacity to dissect the cause-effect relationships linking the activity of a specific protein of interest and the consequent cell behaviors, in time and space. To date, various light-gated protein-interaction modules have been introduced to perturb intracellular protein activities. The one we exploited is based on the interaction between

two proteins from *Arabidopsis Thaliana:* CIB1 and cryptochrome 2 (CRY2) (*Kennedy et al., 2010*). Blue-light illumination induces the heterodimerization of CRY2 with the N-terminus of CIB1 (CIBN). This reaction is reversible and rapid, with response times in the order of seconds (few seconds for dimerization and ~5 min for dissociation after cessation of blue illumination), and does not require exogenous cofactors. In this work, we applied the CRY2-CIBN light dimerization system to selectively activate Ral cascade and to study the primordial phenotypic effect of this activation.

By using this novel optogenetics approach, we precisely established the molecular mechanism underlying the capacity of RalB to drive invasion. This mechanism involves the exocyst-dependent recruitment at the leading edge of the Wave Regulatory Complex (WRC), a five-subunit protein complex involved in the formation of the actin cytoskeleton through interaction with the Arp2/3 complex (*Alekhina et al., 2017*; *Chen et al., 2014*), but unexpectedly independently of the small GTPase Rac1, a well-established WRC activator and master regulator of protrusions (*Ridley et al., 1992*; *Ridley, 2006*). We also found that RalB pathway contribution might be much more relevant than MAPK and PI3K contributions to drive Ras-dependent invasion, as ascertained by using a genetically controlled cell model: the isogenic pair HEK-HT and HEK-HT-H-RasV12 (*Hahn et al., 1999*; *O'Hayer and Counter, 2006*). Light-induced Ral activation was instructive in promoting cell invasion of the non-transformed HEK-HT cells. Finally, we analyzed Ral proteins' expression in a cohort of breast cancer samples, pointing out for the first time a potential role of RalB in the invasiveness and metastatic spread of human breast cancers.

## Results

### Optogenetic control for selective activation of ral proteins

We exploited the CRY2/CIBN light-gated dimerization system (*Kennedy et al., 2010*) to induce activation of endogenous RalA and RalB proteins with a spatial and temporal control. We chose to activate Ral at the plasma-membrane because Ral oncogenic signaling emanates at least in part from the plasma-membrane (*Ward et al., 2001*; *Hamad et al., 2002*; *Lim et al., 2005*). To do so, the GFP-fused CIBN protein was constitutively targeted to the plasma membrane via a K-Ras CAAX motif. The minimal GEF domain of RGL2 (1–518 aa), which is catalytically active on both RalA and RalB (*Ferro et al., 2008*), was fused to CRY2-mCherry (RalGEF-CRY2-mCherry). We stably expressed these two constructs in HEK-HT cells, which are immortalized but not transformed (*Hahn et al., 1999*; *O'Hayer and Counter, 2006*), to generate the 'OptoRal' cell line (CIBN-CAAX/RalGEF-CRY2). As control, we generated the 'OptoControl' cell line which expresses CRY2-mCherry only, without the RalGEF domain (*Figure 1—figure supplement 1*). Upon blue light illumination (100 ms pulses every 15 s), RalGEF-CRY2-mCherry reversibly translocated to the plasma membrane following its binding to GFP-CIBN-CAAX (*Figure 1A*), as shown by TIRF microscopy (*Figure 1B* and *Video 1*). Fluorescence quantifications inside the illuminated area showed that RalGEF-CRY2 recruitment starts in less than 15 s, as expected (*Valon et al., 2015*), reaching a threefold increase in few minutes (*Figure 1C*).

In order to assess whether the recruitment of RalGEF-CRY2 was converted in an efficient Ral activation, we used a Ral activity reporter: the iRFP-tagged Ral GTPase Binding Domain of Sec5 (Sec5GBD), which specifically interacts with both RalA and RalB in their GTP loaded forms (*Takaya et al., 2004*). Since the Sec5GBD-iRFP reporter proteins were in large excess with respect to the endogenous Ral proteins, we overexpressed GFP-RalB in OptoRal cells in order to increase the RalB:Sec5GBD stoichiometric ratio and to achieve the recruitment of a major, detectable fraction of Sec5GBD molecules. Upon blue light illumination, Sec5GBD-iRFP fluorescence at plasma membrane increased 1.4 fold (*Figure 1D,E* and *Video 2*), indicating an efficient activation of Ral proteins, presumably a combination of exogenous GFP-RalB, and endogenous RalA and RalB. The light-induced RalB activation was confirmed by pull-down assay (*Figure 2—figure supplement 1A*) and by a FRET-based RalB activity biosensor (*Figure 2—figure supplement 1B*).

The activation was selective for Ral without cross activation of Rac1 or Cdc42, as shown by overexpressing GFP-Rac1 or GFP-Cdc42 together with a Rac1/Cdc42 activity reporter: the iRFP-tagged GTPase Binding Domain of Pak1 (Pak1GBD) (*Valon et al., 2015*), which specifically interacts with Rac1-GTP and Cdc42-GTP (*Sells et al., 1997*; *Huang et al., 2013*). Upon blue light illumination, Pak1GBD-iRFP fluorescence at plasma membrane did not increase, on the contrary slightly

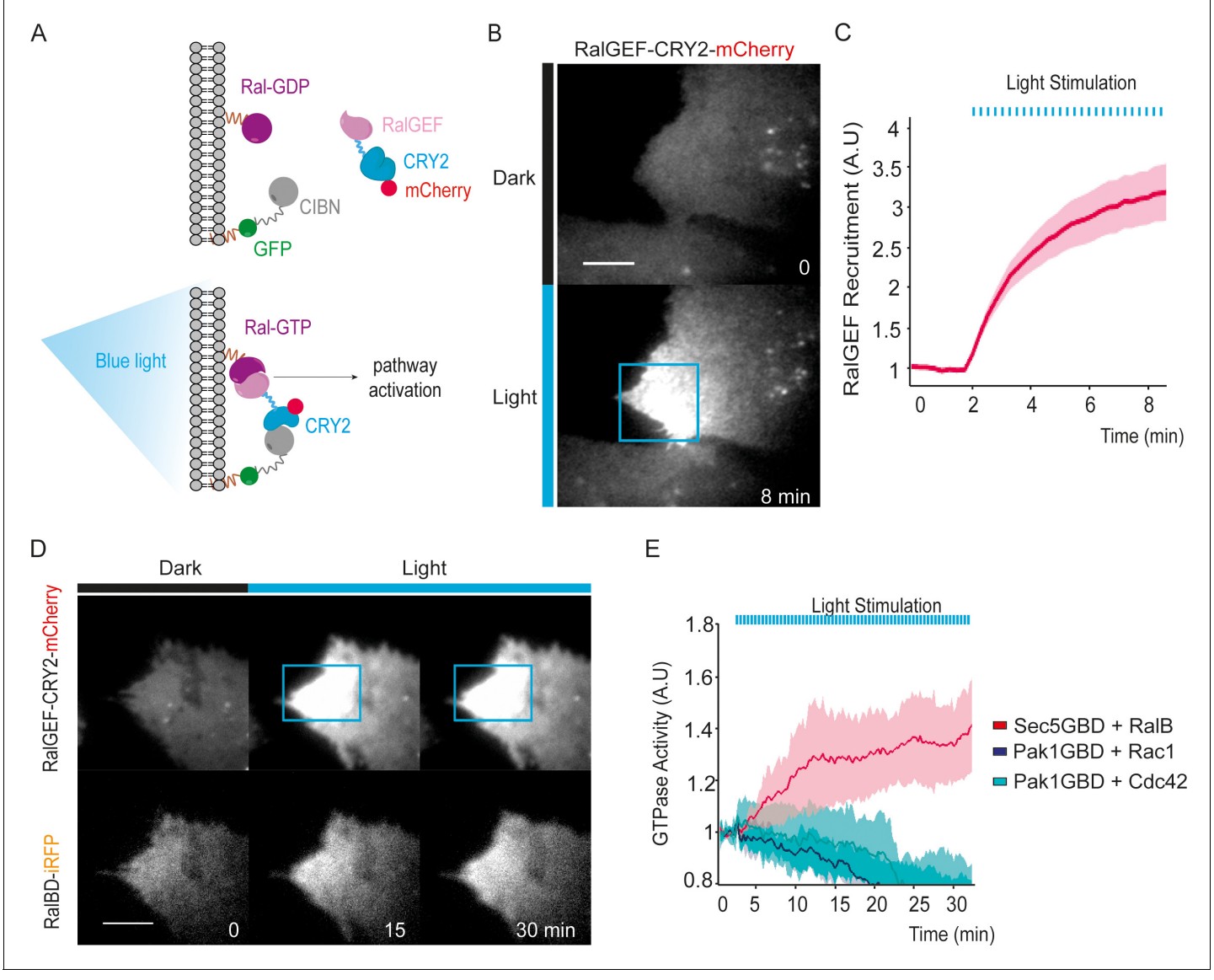

**Figure 1.** Optogenetic control of Ral activation. (**A**) The OptoRal strategy. Upon blue light stimulation the RalGEF domain of RGL2, fused to CRY2, is recruited to the plasma membrane following the interaction between CRY2 and CIBN, which is targeted to the plasma membrane by a CAAX prenylation motif. mCherry and GFP fluorescent proteins were used to monitor expression and localization of RalGEF-CRY2 and CIBN, respectively. After recruitment, the RalGEF induces activation of endogenous Ral. (**B**) Representative RalGEF-CRY2-mCherry recruitment. The fluorescent RalGEF-CRY2-mCherry fusion protein was imaged by TIRF microscopy before illumination (dark) and 8 min after blue light stimulation inside the blue square area (100 ms pulses every 15 s). Scale bar, 10 μm. See *Video 1* for the entire sequence. (**C**) Quantification of RalGEF-CRY2-mCherry recruitment. Average time course of the fold increase of mCherry fluorescence, that is RalGEF recruitment, inside the illuminated square area, is calculated from n = 20 cells from three independent experiments. Lines represent the mean, shaded regions represent the standard deviation (SD). (**D**) Representative Ral activation. The fluorescent RalGEF-CRY2-mCherry and Sec5GBD-iRFP (reporter of Ral activity) fusion proteins were simultaneously imaged by TIRF microscopy. Scale bar, 10 μm. See *Video 2* for the entire sequence. (**E**) Light activates RalB, but not Rac1 or Cdc42. OptoRal cells were transiently transfected to express: Sec5GBD-iRFP (reporter of Ral activity) with GFP-RalB (red line), Pak1GBD-iRFP (reporter of Rac1/Cdc42 activity) with GFP-Rac1 (blue line) or Pak1GBD-iRFP with GFP-Cdc42, respectively (light blue line). Average time course of the fold increase of iRFP fluorescence, that is Ral or Rac1/Cdc42 activities, inside the illuminated square area, is calculated from n = 6 cells per condition from three independent experiments. Lines represent the mean, shaded regions represent the standard deviation (SD).

DOI: https://doi.org/10.7554/eLife.40474.003

The following figure supplement is available for figure 1:

**Figure supplement 1.** Optogenetic pair expression.

DOI: https://doi.org/10.7554/eLife.40474.004

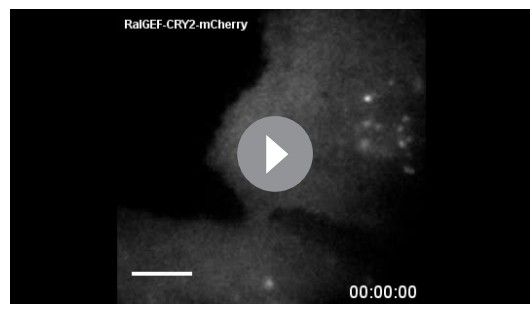

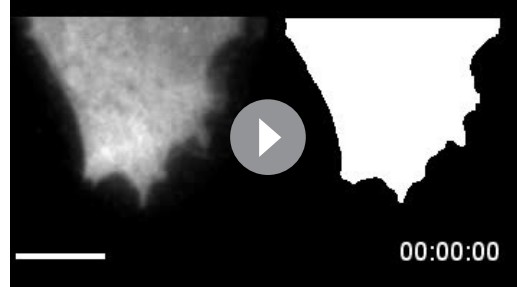

**Video 1.** Blue light induces local recruitment at plasma membrane of RalGEF-CRY2-mCherry in OptoRal cells. OptoRal cells were imaged by TIRF microscopy (Cherry channel). Cells were illuminated inside the blue rectangle starting at time t = 2.15 min. Scale bar is 10 µm. Time length: 8.5 min. Images were acquired every 15 s and the video is shown at 7 frame/s.
DOI: https://doi.org/10.7554/eLife.40474.005

**Video 3.** Local activation of Ral induces protrusions. On the left, optogenetic recruitment of of RalGEF (Cherry channel) in OptoRal cells imaged by TIRF microscopy. On the right, threshold-based binary mask of the cells used to calculate the edge velocity. Cells were illuminated inside the blue rectangle starting at time t = 3.10 min. Scale bar is 10 µm. Time length: 6.83 min. Images were acquired every 10 s and the video is shown at 7 frame/s.
DOI: https://doi.org/10.7554/eLife.40474.010

decreased (*Figure 1E*), indicating the absence of Rac1 or Cdc42 activation upon recruitment of RalGEF at the plasma membrane.

In conclusion, we developed an efficient and selective optogenetic OptoRal system to specifically trigger local activation of Ral proteins. Since the optogenetic perturbations act in a timescale (tens of second) much faster than the ones of endogenous feedbacks (minutes) (*Valon et al., 2015*), this new approach allowed us to investigate the direct consequences of Ral activation, at both phenotypic and molecular levels.

## Local Ral activation triggers protrusions, independently of Rac1

A conspicuous phenotypic consequence of local Ral activation was an increase in cell edge dynamics, leading to protrusion formation (*Video 3*). By using an automated method (*Paul-Gilloteaux et al., 2018*), we tracked over time the cell contour inside the illuminated area and we generated edge velocity maps (*Figure 2A*). The velocity maps of OptoRal cells showed an evident stimulation of cell edge dynamics after light stimulation (*Figure 2B*). This stimulation corresponded, for the majority of the sampled space and time points, to an increase of positive velocity (i.e. protrusions) and not of negative velocities (i.e. retractions). In contrast, no changes of cell edge dynamics were observed upon light stimulation of OptoControl cells (*Figure 2C* and *Video 4*), excluding non-specific effects due to illumination or to CRY2 recruitment. Moreover, the alternations of dark and light periods revealed that the velocity dynamics were reversible and exquisitely controllable by the illumination in OptoRal cells (*Figure 2—figure supplement 2A*) but not in the OptoControl cells (*Figure 2—figure supplement 2B*).

The quantifications of edge velocities from several OptoRal cells before and after illumination showed a highly significant increase of membrane protrusion events, but not of retraction events; this increase was not found in OptoControl cells (*Figure 2D*). To address a possible role

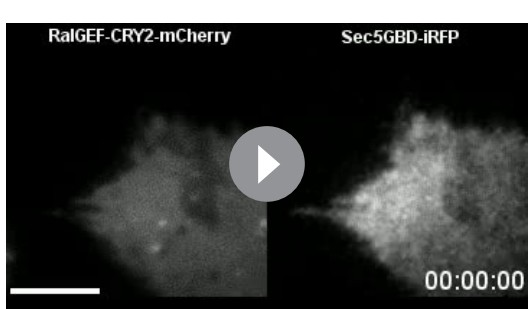

**Video 2.** Blue light induces local Ral activation in OptoRal cells. OptoRal cells expressing RalB-GFP and Sec5GBD-iRFP were imaged by TIRF microscopy. On the left, the recruitment of RalGEF is followed in the mCherry channel. On the right, the recruitment of Sec5GBD (reporter for Ral activity) is followed in the iRFP channel. Cells were illuminated inside the blue rectangle starting at time t = 2 min. Scale bar is 10 µm. Time length: 32.25 min. Images were acquired every 15 s and the video is shown at 7 frame/s.
DOI: https://doi.org/10.7554/eLife.40474.006

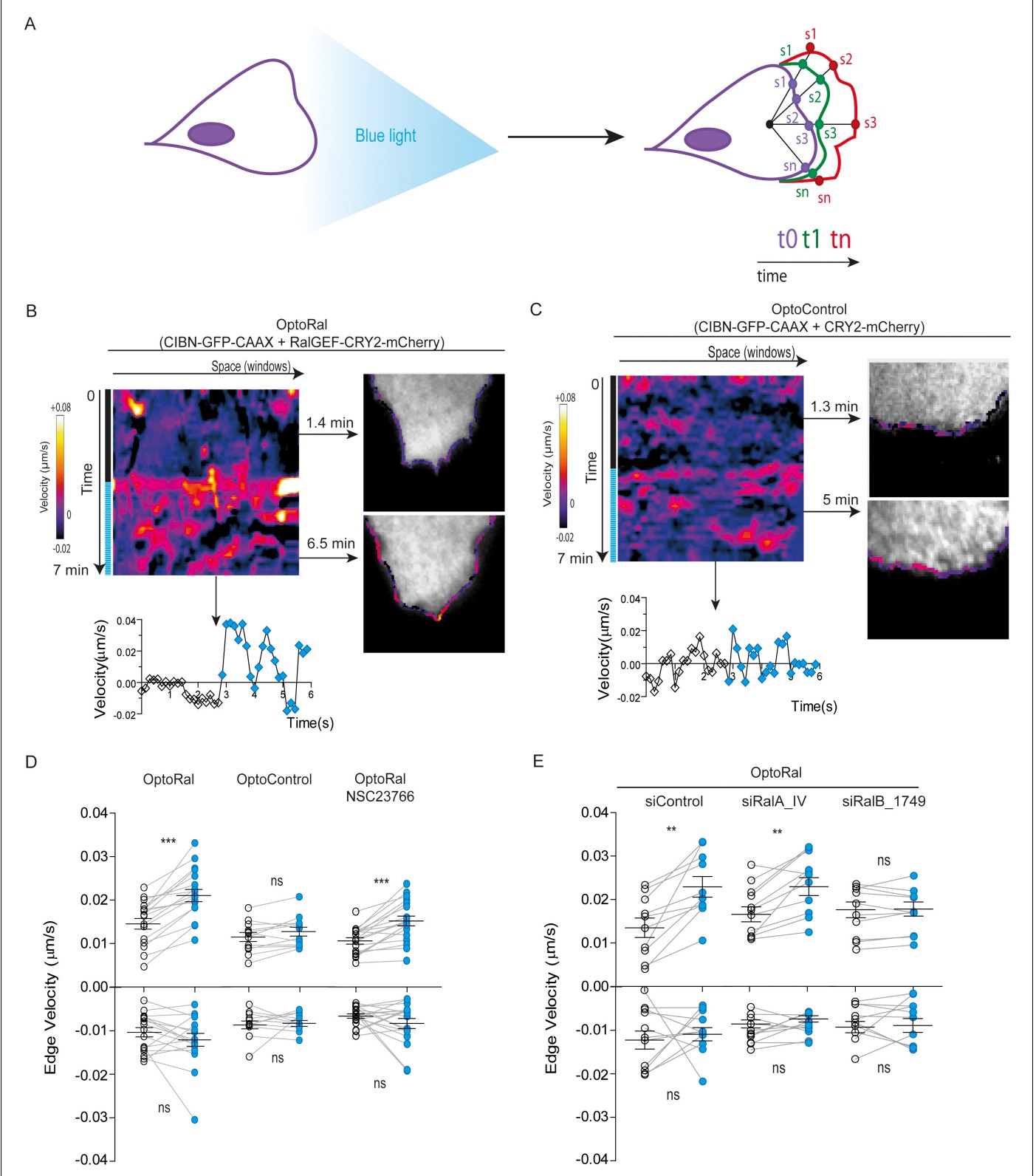

**Figure 2.** Local Ral activation induces protrusiveness. (**A**) Cell edge dynamics inside the illuminated area were measured using an automated method. (**B**) Edge dynamics velocity map of a representative OptoRal cell before and after illumination (dotted blue line). See *Video 3* for the entire sequence. The color-coded map shows the velocity measurements for each sampled edge window (space, horizontal axis) and for each time point (time, vertical axis). Positive velocities (i.e. protrusion) are represented as warm color, negative velocities (i.e. retraction) as cold colors. As example of velocity profile

*Figure 2 continued on next page*

*Figure 2 continued*

over time, the velocity measurements are shown for a selected point in space (window 71) in the lower panel; the stimulation of edge velocity is visible immediately after light illumination. As examples of velocity profiles over the space dimension, the velocity measurements are shown in the right panels using the color-code along the cell edge for two selected time points, 1.4 min and 6.5 min, dark and light conditions, respectively. (C) Edge dynamics velocity map of a representative OptoControl cell before and after illumination. See *Video 4* for the entire sequence. The velocity profiles over time (window 77, lower panel) or over space (1.3 min and 5 min, right panels) show that illumination did not induce any change of edge velocity. (D) Quantification of edge velocities before and after illumination. Positive velocities (i.e. protrusion) and negative velocities (i.e. retraction) are analysed separately. Each white dot represents the mean of all velocity measurements (all time and space points) before illumination. Each blue dot represents the mean of all velocity measurements (all time and space points) after illumination. Measurements from a same cell are connected by lines. Illumination stimulates positive edge velocities in OptoRal cells (left) but not in OptoControl cells (centre). Inhibition of Rac with NSC23766 (100 μM, treatment started 1 hr before experiment) did not impair stimulation by light of positive edge velocities in OptoRal cells (right). Bars represent mean of n = 18 cells per condition ±SEM from three independent experiments. *** indicates p<0.001, ** indicates p<0.01 and ns indicates not-significant, using Wilcoxon signed-rank test for paired measurements (same cell pre- and post-illumination). There was no significant difference in the delta velocities pre- and post-illumination when comparing presence and absence of the NSC23766 inhibitor (using Student t-test, not shown). There was no significant difference in the pre-illumination edge velocities when comparing OptoRal and OptoControl cells (using Student t-test, not shown). (E) RalB depletion, but not RalA depletion, impairs stimulation of positive edge velocities in OptoRal cells. OptoRal cells were transfected with the indicated siRNAs and analysed as in panel D. n = 10–11 cells per condition.

DOI: https://doi.org/10.7554/eLife.40474.007

The following figure supplements are available for figure 2:

**Figure supplement 1.** Additional evidences that light stimulates RalB activity in OptoRal cells.
DOI: https://doi.org/10.7554/eLife.40474.008

**Figure supplement 2.** Reversibility and repeatability of edge velocity stimulation by light. Validation of Rac1 inhibition and of RalA and RalB depletion.
DOI: https://doi.org/10.7554/eLife.40474.009

of Rac in this phenotype we treated the OptoRal cells with the RacGEF inhibitor NSC23766. Treatment with NSC23766 (100 μM, 1 hr) substantially reduced (~60%) Rac1-GTP level (*Figure 2—figure supplement 2C*), but it did not impair a fully efficient protrusion induction by illumination in OptoRal cells (*Figure 2D*), suggesting that Rac1 have a marginal role in stimulation of edge velocities when Ral proteins are activated. Importantly, blue illumination of OptoRal cells does not activate at all Rac1, as shown by the Pak1GBD fluorescent reporter (*Figure 1E*). These results together support the conclusion that Ral triggers protrusion independently of Rac1.

Our OptoRal system is designed to activate both endogenous RalA and RalB. However, RalB-depleted cells, but not RalA depleted cells, were impaired in protrusions formation, proving that light-controlled stimulation of cell protrusions is essentially due to activation of endogenous RalB, rather than RalA (*Figure 2E* and *Figure 2—figure supplement 2D* for depletion validation), consistently with the well-established specific role of RalB in the regulation of motility and invasion (see Introduction).

## Upon Ral activation the WRC complex is recruited at the cell front via its association to the exocyst complex

Next, we addressed the question of the molecular mechanisms underlying the capacity of RalB to trigger protrusions independently of Rac1. A previous work in our lab (*Biondini et al., 2016*) established the existence of an interaction between the Wave Regulatory Complex (WRC), a crucial regulator of actin polymerization and protrusion formations (*Alekhina et al., 2017*; *Chen et al., 2014*), and the exocyst complex (*Wu and Guo, 2015*), a major direct effector of Ral, but the functional consequences of this interaction are still unclear. We reasoned that Ral activation might promote the trafficking and recruitment of the exocyst/WRC complex assembly at the leading edge (*Figure 3A*). To test this hypothesis, we generated OptoRal and OptoControl cell lines stably expressing the iRFP-fused Abi1 subunit of WRC complex. The correct incorporation of iRFP-Abi1 into the WRC complex was verified by size exclusion chromatography: the exogenous iRFP-Abi1, as well as the endogenous Abi1, co-eluted with the Cyfip subunit in fractions of approximately 400 kDa corresponding to the size of the whole WRC complex (*Gautreau et al., 2004*) (*Figure 3—figure supplement 1A,B*).

We locally activated Ral at cell periphery and followed iRFP-Abi1 recruitment. For convenience of imaging, we limited the inter-cellular morphology variability by subjecting the OptoRal cells to wound healing, leading to the formation of large front lamellipodia where Ral was activated by light.

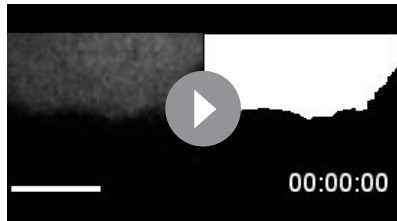

**Video 4.** Light stimulation of OptoControl cells does not induce protrusions. Same conditions as *Video 3*.
DOI: https://doi.org/10.7554/eLife.40474.011

TIRF images showed that Ral activation induced an immediate increase of iRFP-Abi1 signal at the plasma membrane, both at the leading edge and at the ventral side (*Figure 3B* snapshots and *Video 5*). In contrast, OptoControl cells did not show any light-dependent iRFP-Abi1 recruitment (*Figure 3C* snapshots and *Video 6*). WRC edge recruitment was quantified by using an automated method that measures the maximum fluorescent intensity at the edge (width of 1.12 μm) (*Paul-Gilloteaux et al., 2018*): the resulting heat maps show a dynamic edge fluorescence appearing upon illumination in OptoRal cells (*Figure 3D*, left) but not in OptoControl cells (*Figure 3D*, right). We could not apply this method of quantification at the edge on all the imaged cells, because the leading edge was often lifted above the ventral plan captured by TIRF and because the cycling dynamics (protrusion-retraction alternation and on-off Abi1 recruitment) were very fast. Instead, fluorescence quantifications from several OptoRal and OptoControl cells could be easily done at ventral location inside the illuminated area: iRFP-Abi1 was recruited in OptoRal cells but not in OptoControl cells, with a robust statistical significance (*Figure 3E*).

To demonstrate that WRC translocation is a direct consequence of its association with the exocyst, we exploited a previously characterized loss-of-interaction Abi1 mutant, Abi1Q56A, which is specifically impaired in its binding to the Exo70 subunit of exocyst (*Biondini et al., 2016*). We generated an OptoRal cell line expressing the iRFP-Abi1Q56A mutant and we verified its correct incorporation into WRC by size exclusion chromatography (*Figure 3—figure supplement 1C*). The analysis of several cells showed that upon Ral activation the recruitment at the ventral plasma membrane of the AbiQ56A mutant was significantly and substantially reduced with respect to wild-type Abi1 (*Figure 3E*), indicating that the WRC/exocyst association is required for efficient translocation of WRC complex. Moreover, while illumination stimulated edge velocity of OptoRal cells expressing Abi1 wild-type, it was not effective in stimulating edge velocity of OptoRal cells expressing the Abi1Q56A mutant (*Figure 3F*), indicating that reduction of WRC recruitment correlated with a decrease in protrusion dynamics.

## Activation of Ral-exocyst-WRC axis promotes invasion of non-transformed cells

In order to functionally assess the impact on invasion of light-controlled Ral activation, we performed a Transwell invasion assay coupled with optogenetic illumination. For this purpose, we designed and built a customized 12-well array LED to deliver optical stimuli to the cells inside the Transwells (*Figure 4—figure supplement 1*). Blue light was administrated continuously from the bottom of the Transwell plate for the whole duration of the invasion assay (6 hr). Using this device, we compared the invasive capabilities of OptoControl and OptoRal cells, as well of OptoControl iRFP-Abi1 wild-type, OptoRal iRFP-Abi1 wild-type, and OptoRal iRFP-Abi1Q56A, in both dark and light stimulation conditions.

Illumination of OptoRal cells, but not of OptoControl cells, was sufficient to induce a substantial increase of invasive capabilities (*Figure 4A*). Similar results were obtained upon expression of the wild-type Abi1 subunit of WRC complex, whereas OptoRal cells expressing Abi1Q56A (the allele specifically impaired in its binding to exocyst [*Biondini et al., 2016*]) failed to invade upon light stimulation (*Figure 4B*). These results strongly support the conclusion that activation of Ral drives cell invasion by recruiting WRC via the exocyst.

## Ras-driven cancer invasion requires activation of RalB via RGL1 and RGL2

To evaluate the contribution of Ral activation downstream oncogenic Ras in cell invasion, we took advantage of the fact that HEK-HT cells, non-transformed, develop a tumorigenic, invasive and metastatic phenotype in vivo upon expression of oncogenic H-RasV12 (*Hahn et al., 1999*; *O'Hayer and*

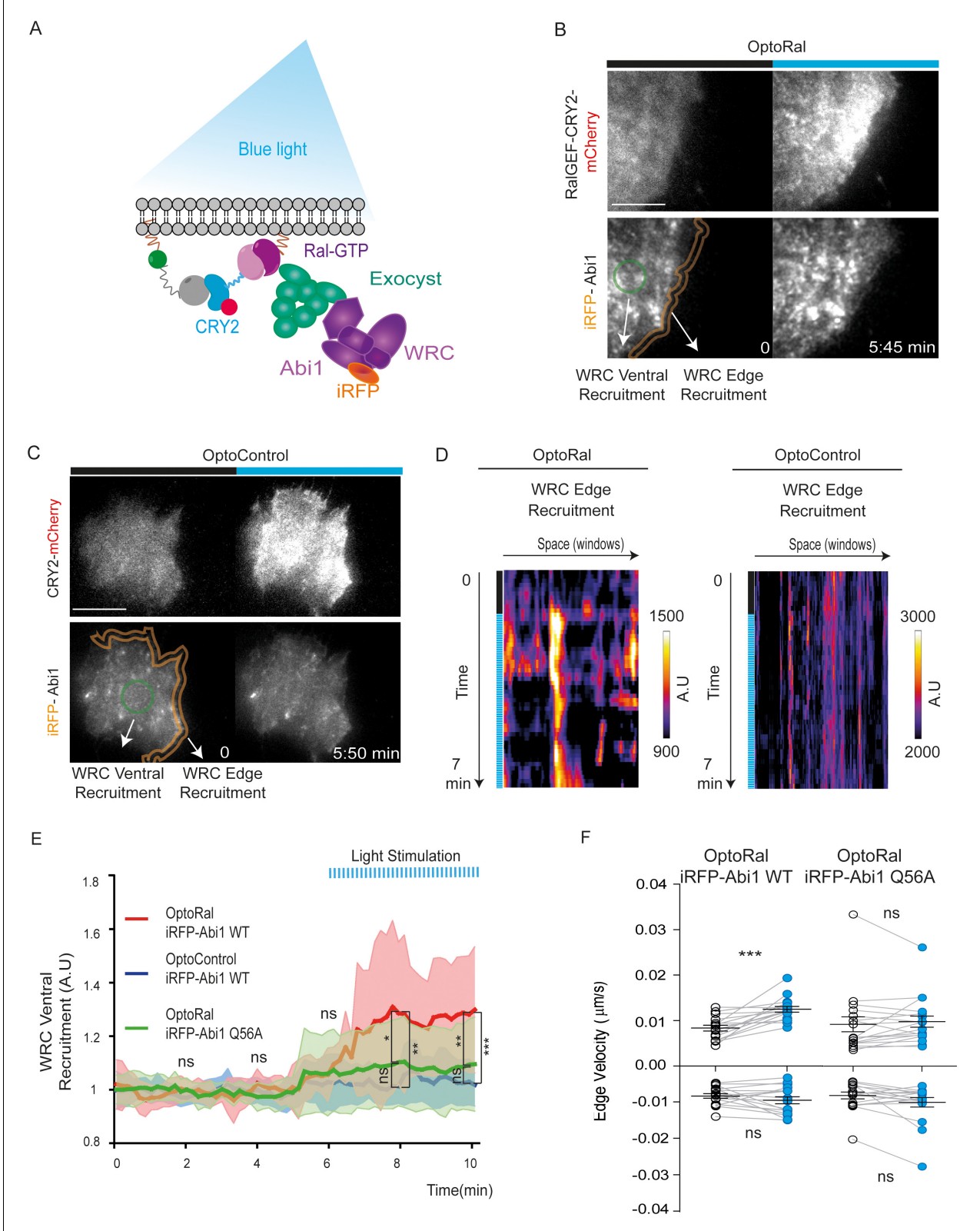

**Figure 3.** Local Ral activation induces WRC complex recruitment at the edge and ventral plasmamembrane via the exocyst. (**A**) Working model. Upon Ral activation, exocyst is assembled and recruited at the plasma membrane, functioning as transporter for the WRC complex. The model was tested by monitoring the recruitment of WRC by TIRF microscopy using a fluorescent-tagged Abi1 subunit of WRC (iRFP-Abi1). (**B**) Representative iRFP-Abi1 recruitment (reporter of WRC localization) in OptoRal cells. The fluorescent RalGEF-CRY2-mCherry and iRFP-Abi1 fusion proteins were simultaneously

*Figure 3 continued on next page*

*Figure 3 continued*

imaged with TIRF microscopy before and 7 min after blue light stimulation. Scale bar, 10 µm. See *Video 5* for the entire sequence. The variations of iRFP fluorescence were measured both at the ventral compartment (green circle) and at the leading edge (orange lines) of the cells. (C) Representative absence of light-dependent iRFP-Abi1 recruitment in OptoControl cells. Scale bar, 10 µm. See *Video 6*. There is an increase of mCherry but not of iRFP fluorescence after illumination. (D) Color-coded maps show the WRC edge recruitment in the OptoRal cell of panel B and in the OptoControl cell of panel C, calculated across a band with a width of 1.12 µm, before and after illumination, for each sampled edge window (space, horizontal axis) and for each time point (time, vertical axis). In the OptoRal cell (left), changes to warmer colors after illumination indicate increased WRC recruitment. In the OptoControl cell (right), there are no changes of WRC edge recruitment after illumination. (E) Quantification of ventral iRFP-Abi1 recruitment: comparison of OptoRal iRFP-Abi1 wild-type (WT), OptoControl iRFP-Abi1 wild-type (WT), and OptoRal iRFP-Abi1Q56A cells. Average time course of the fold increase of iRFP fluorescence, that is WRC recruitment, is calculated from n = 15 cells per condition from four independent experiments. Lines represent the mean, shaded regions represent the standard deviation (SD). Statistical comparison of the three curves was done using Student t-test at 2, 4, 6, 8, 10 min time points: *p<0.05, **p<0.01, ***, p<0.001, ns not-significant. Only the OptoRal iRFP-Abi1 wild-type curve diverges after light stimulation (at 6 min). (F) Quantification of edge velocities before and after illumination: comparison Abi1 wild-type (WT) versus Abi1Q56A in OptoRal cells. See legend *Figure 2D* for details. Illumination stimulates positive edge velocities in OptoRal cells expressing Abi1 WT (left) but not in OptoRal cells expressing Abi1Q56A (right). Bars represent mean of n = 15 cells per condition ±SEM from three independent experiments. *** indicates p<0.001 and ns indicates not-significant, using Wilcoxon signed-rank test for paired measurements (same cell pre- and post-illumination).

DOI: https://doi.org/10.7554/eLife.40474.012

The following figure supplement is available for figure 3:

**Figure supplement 1.** Size exclusion chromatography shows incorporation of exogenous iRFP-Abi1 (both wild-type and Q56A) into the whole WRC complex.

DOI: https://doi.org/10.7554/eLife.40474.013

*Counter, 2006*). We compared this isogenic pair, HEK-HT and HEK-HTRasV12, by using two different in vitro invasion assays: the widely used 'Transwell Invasion assay' (invasion through a thin matrigel layer) and the 'Inverted Invasion assay' with a more 3D setting (invasion through a thick collagen gel). In both assays, we found as expected that HEK-HT cells did not invade at all, while HEK-HT-RasV12 displayed a strong invasive capacity (*Figure 5A,B*). Interestingly, the HEK-HT-RasV12 cells invaded as multi-cellular clusters in the 3D collagen gel (*Figure 5B*, photos below).

In HEK-HTRasV12 cells, the silencing of RalB impaired Transwell invasion of approximatively 60%, the silencing of RalA had not effect, and the silencing of both RalB and RalA reached up to 90% invasion inhibition (*Figure 5C*) (see *Figure 5—figure supplement 1A* for depletion efficiencies), indicating that Ras-dependent RalB activation substantially contributes to the invasive phenotype of Ras-mutated cells, and that RalA is dispensable, but it might partially compensate when RalB is absent. The specific role of RalB, with respect to RalA, for cell motility and invasion is in perfect agreement with several previous works (*Oxford et al., 2005*; *Rossé et al., 2006*; *Lim et al., 2006*; *Rybko et al., 2011*). The RalB requirement for in vitro invasion of HEK-HTRasV12 cells was confirmed using the Inverted Invasion assay with collagen (*Figure 5D*) (see *Figure 5—figure supplement 1B* for depletion efficiency up to 5 days post-siRNA transfection).

The double mutant H-RasV12G37, which is completely impaired in stimulating MAP kinase activity but maintains Ral activation (*White et al., 1995*; *Bettoun et al., 2016*) (*Figure 5—figure supplement 1C*), showed the same invasive capabilities by Transwell invasion assay as compared with H-RasV12 (*Figure 5E*, center), indicating that Ras-dependent MAPK hyper-activation is dispensable to drive efficient invasion, in this genetically controlled cell model. Since HEK-HT-RasV12G37 cells retained a residual Ras-dependent activation of PI3K pathway activation, as assessed by AKT phosphorylation (*Lim et al., 2005*; *Bettoun et al., 2016*), we used a PI3K

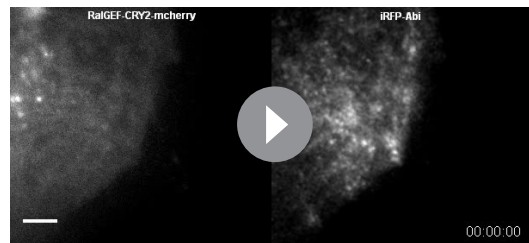

**Video 5.** Local activation of Ral induces WRC recruitment at plasma membrane. OptoRal cells expressing iRFP-Abi WT were imaged by TIRF microscopy. On the left, the recruitment of RalGEF is followed in the mCherry channel. On the right, the recruitment of Abi1 (a subunit of WRC complex) is followed in the iRFP channel. Cells were illuminated inside the blue rectangle starting at t = 5 min. Scale bar is 10 µm. Time length: 10.17 min. Images were acquired every 10 s and the videos are shown at 7 frame/s.

DOI: https://doi.org/10.7554/eLife.40474.014

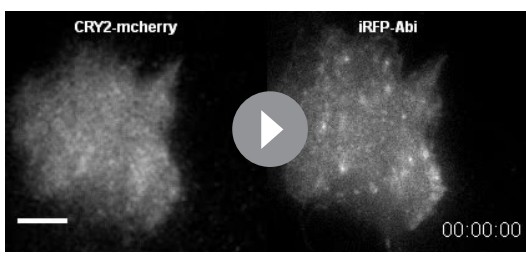

**Video 6.** Light stimulation of OptoControl cells does not induce WRC recruitment at plasma membrane. Same conditions as *Video 5*.
DOI: https://doi.org/10.7554/eLife.40474.015

inhibitor (PIK90), which targets all PI3K isoforms, to completely inhibit PI3K pathway (*Figure 5— figure supplement 1D*). PIK90-treated HEK-HT-RasV12G37 cells are still capable to invade as efficiently as untreated HEK-HT-RasV12G37 and HEK-HT-RasV12 cells (*Figure 5E*, right), indicating that also Ras-dependent PI3K hyper-activation is dispensable to drive efficient Transwell invasion.

These unexpected results were confirmed with HEK-HT-RasV12 cells by a purely pharmacological approach using the MEK inhibitor trametinib and the PI3K inhibitor PIK90, alone or in combination. Despite a nearly complete inhibition of Erk or Akt phosphorylation, trametinib-treated or PIK90-treated HEK-HT-RasV12 cells were still able to invade to the same extent as control vehicle-treated cells; only the combined MAPK and PI3K blockage displayed an inhibitory tendency, although not statistically significant (*Figure 5F*).

Taken together these results showed that the Ras-RalGEF-RalB signaling axis is necessary to promote invasion in Ras-transformed cells, while MAPK and PI3K pathways appear to be dispensable, at least in our experimental context.

Next, we aimed at identifying the missing molecular link between Ras and RalB, that is the specific RalGEFs implicated in Ras-dependent cell invasion. We thus silenced each of the six RalGEFs by two independent siRNA in HEK-HT-RasV12 cells. The depletion of RGL1 and RGL2 substantially reduced invasion. RLGL3, RalGDS and RalGPS1 were clearly not required for invasion, while RalGPS2 silencing showed a potential inhibitory effect, even though statistically not significant (*Figure 6A*). RGL1 and RGL2 have a Ras-association domain (RA). We therefore concluded that RGL1 and RGL2 are the molecular actors that promote invasion down-stream oncogenic Ras by activating RalB.

By immunofluorescence staining, we found that endogenous RGL2 (*Figure 6B*) and RalB (*Figure 6C*) were more recruited at cell edges in HEK-HTRasV12 cells with respect to normal HEK-

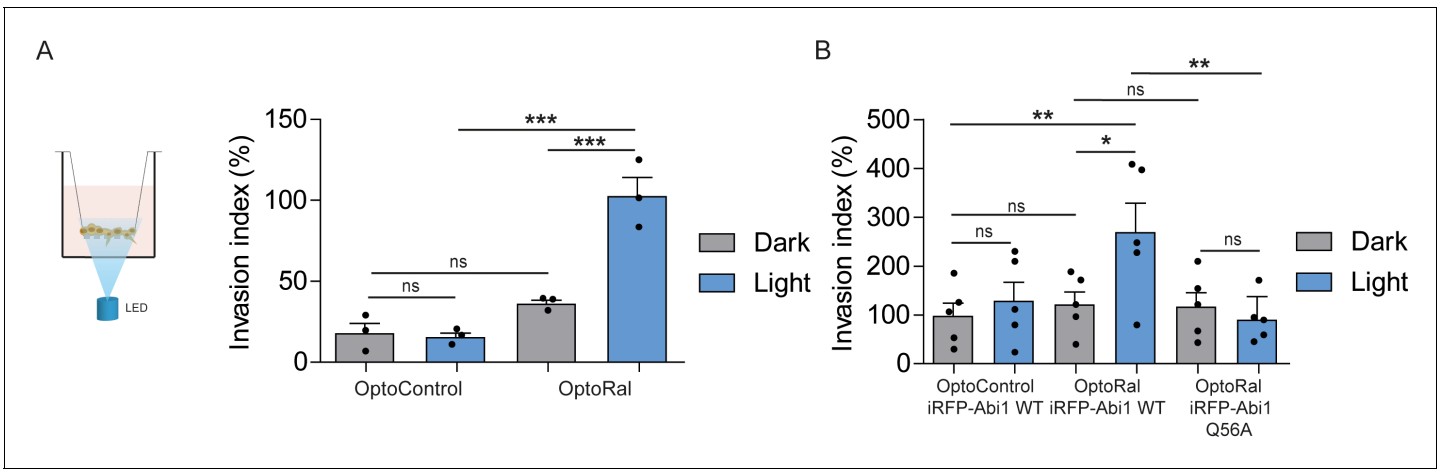

**Figure 4.** Activation of Ral triggers invasion of non-transformed cells. (**A**) Transwell invasion assay with OptoControl and OptoRal cells, in the dark or under light stimulation. Graph shows the mean ±SEM from three independent experiments. In the graph the results of one-way ANOVA tests are reported to compare pairs of conditions. (**B**) Transwell invasion assay with OptoControl iRFP-Abi1 wild-type, OptoRal iRFP-Abi1 wild-type, and OptoRal iRFP-Abi1Q56A cells, in the dark or under light stimulation. Graph shows the mean ±SEM from five independent experiments. In the graph the results of one-way ANOVA tests are reported to compare pairs of conditions. *$p<0.05$, **$p<0.01$, ***, $p<0.001$, ns not-significant.
DOI: https://doi.org/10.7554/eLife.40474.016

The following figure supplement is available for figure 4:

**Figure supplement 1.** The custom-made illumination device for optogenetic activation on multi-well dishes.
DOI: https://doi.org/10.7554/eLife.40474.017

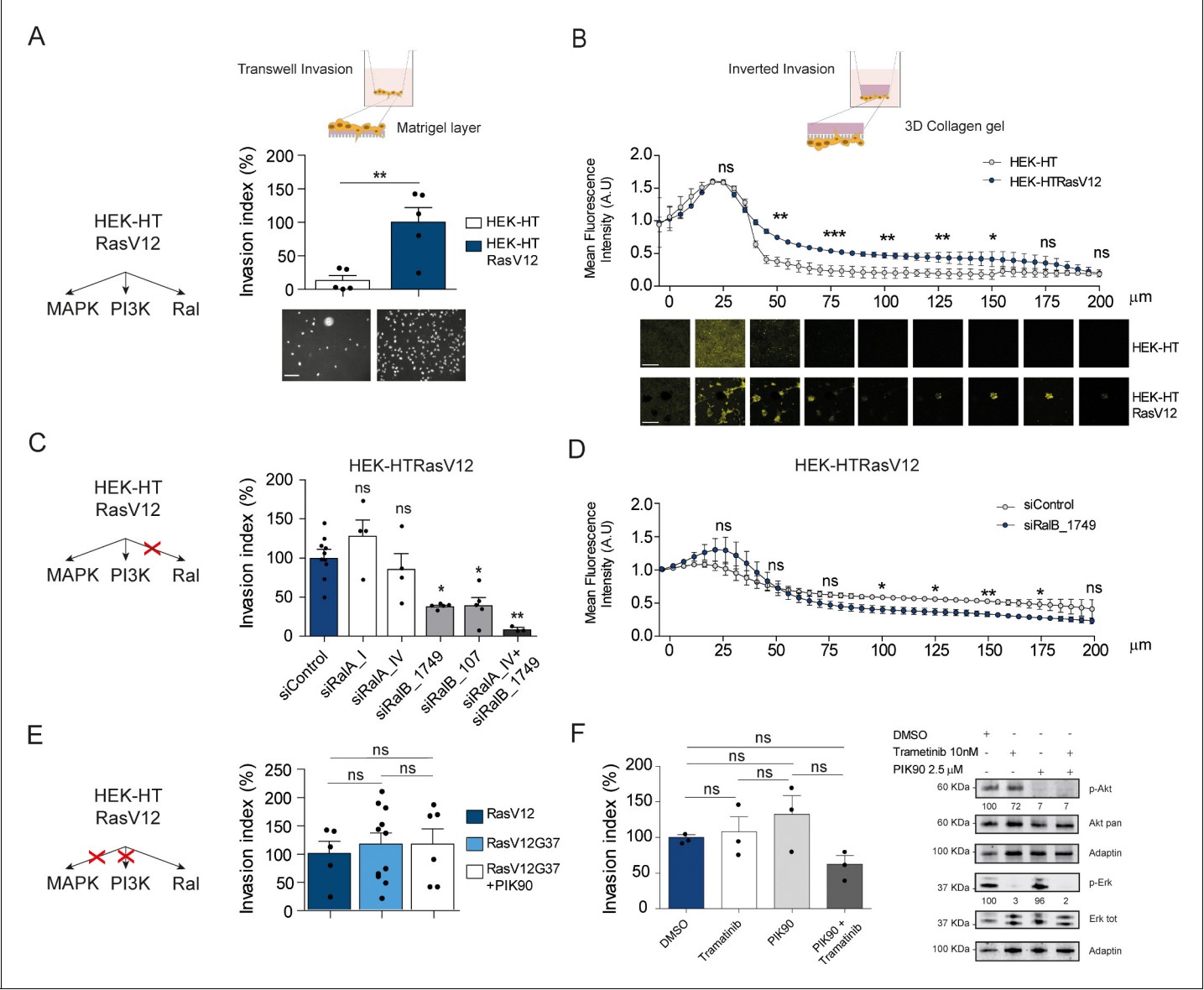

**Figure 5.** Ras-driven invasion requires RalB, rather than MAPK and PI3K. (**A**) Comparison of invasive capability of HEK-HT and HEK-HT-H-RasV12 cells through a thin matrigel layer using Transwell invasion assay (6 hr). Representative images of invading cells stained with DAPI are shown below the graph. Scale bar, 50 µm. Graph shows the mean ±SEM. Each dot represents one experiment. For statistics Student t-test was used. (**B**) Comparison of invasive capability of HEK-HT and HEK-HT-HRasV12 cells through a 3D bovine collagen gel (2.3 mg/ml) using Inverted invasion assay (4 days). Cells were live stained with CellTrace. Invasion was quantified as mean fluorescence of confocal fields acquired every 5 µm starting from below the porous membrane till 200 µm distance inside the gel. Graph shows the mean ±SD of measurements in duplicate from three experiments. Representative images are shown below the graph. Scale bar, 250 µm. Statistical comparison of the curves was done using Student t-test at every 25 µm. (**C**) RalB but not RalA is required for HEK-HTRasV12 invasion through a matrigel layer (Transwell invasion assay). Cells were transfected with the indicated siRNAs. For statistics one-way ANOVA test was used. (**D**) RalB is required for HEK-HTRasV12 invasion through a 3D collagen gel (Inverted invasion assay). Invasion was quantified as in panel B. (**E**) Comparison of invasive capability of HEK-HT-H-RasV12, HEK-HT-H-RasV12G37 and PIK90-treated HEK-HT-H-RasV12G37 cell lines using Transwell invasion assay. The combination of the additional G37 mutation and treatment with the PI3K inhibitor PIK90 (10 µM, treatment started 1 hr before Transwell invasion assay and maintained during the 6 hr invasion) impairs RasV12-dependent MAPK and PI3K hyperactivation, conserving only the activation of Ral pathway. For statistics one-way ANOVA test was used. (**F**) Comparison of invasive capability of HEK-HT-H-RasV12 cells treated with DMSO, with the MEK inhibitor Trametinib (10 nM), with the PI3K inhibitor PIK90 (2.5 µM), or with both Trametinib and PIK90. Treatments were started 1 hr before Transwell invasion assay and maintained during the 6 hr invasion. Cell lysates were prepared at 1 hr post-treatment. On the right, representative western blots for total Akt and phospho-Akt (Ser437), and for total Erk and phospho-Erk (Thr202/Tyr204), show the efficient pharmacological blockage of MAPK and PI3K pathways. Quantifications of Akt and Erk phosphorylation, calculated as p-Akt/total Akt or p-Erk/total Erk ratios, and normalized for HEK-HT-HRasV12 without drugs, are shown below the WBs. For statistics one-way ANOVA test was used. *p<0.05, **p<0.01, ***p<0.001, ns not-significant.

*Figure 5 continued on next page*

*Figure 5 continued*

DOI: https://doi.org/10.7554/eLife.40474.018

The following figure supplement is available for figure 5:

**Figure supplement 1.** Representative western blots.

DOI: https://doi.org/10.7554/eLife.40474.019

HT cells, consistent with the model in which plasma-membrane Ras-GTP binds and recruits RGL2, which in turn binds and activates its substrate RalB, promoting invasiveness via exocyst-mediated WRC recruitment (*Figure 6D*). Noteworthy, both RGL2 and RalB localize also at endomembranes, where local signaling might drive additional outputs.

## RalB protein expression increases in a manner consistent with disease progression in human breast cancers

While Ral proteins have been involved in several cancers frequently carrying Ras mutations, such as pancreas, lung, colon, bladder, melanoma (*Yan and Theodorescu, 2018*), their roles have not been explored in breast cancers, in which the Ras mutation frequency is rather low (2.3%) (cbioportal) (*Gao et al., 2013*). We investigated by immunohistocytochemistry (IHC) the protein levels of RalA and RalB in 502 invasive ductal carcinoma representative of the four main molecular subtypes (luminal A, luminal B, HER2+, triple-negative) (see *Figure 7—figure supplement 1A* for intensity scores). In invasive ductal carcinoma, RalB protein expression was slightly higher in luminal A and B than in HER2 +and triple-negative (TN) tumors (*Figure 7—figure supplement 1B*), while RalA protein expression was slightly lower in TN tumors as compared to the other subtypes (*Figure 7—figure supplement 1C*). More interestingly, both RalB and RalA expression was higher in tumor cells than in normal juxtatumoral cells (*Figure 7A,B*). We also relatively compared the Ral staining at in situ and invasive compartments of invasive ductal carcinoma, and at lymph node metastasis. Strikingly, RalB expression significantly increased in a manner consistent with disease progression: the median H-score was = 0.5 in normal juxtatumoral cells,=1 in tumor cells of in situ compartment,=1.5 in tumor cells of invasive compartment, and = 2 in tumor cells at lymph node metastasis (*Figure 7A,C*). On the contrary, RalA expression did not change in tumor cells regardless of the in situ, invasive or metastatic localization (*Figure 7B*). With the limitation that RalB amount does not necessarily reflect RalB activity, these results suggest that RalB might have a role in human breast cancer invasion and metastasis. Consistent with this hypothesis, RalB silencing was able to impair invasion (by Transwell invasion assay) of two triple-negative breast cancer cell lines: MDA-MB-231 (carrying the KRasG13D mutation) and BT549 (Ras wild-type) (*Figure 7D* and *Figure 7—figure supplement 1D*).

## Discussion

Collectively our findings established the utter relevance of the activation of RalB to promote invasion. By exploiting optogenetics to study causality, we showed that not only RalB was permissive for protrusion formation and invasion, but it was instructive in absence of any oncogenic mutation. Moreover, we provide a detailed molecular model of how the Ras-RalB signaling axis governs cell invasion (*Figure 6D*): active Ras binds and recruits the two RalGEFs RGL1 and RGL2, which activate RalB; activated RalB binds to the exocyst complex, promoting its assembly (*Moskalenko et al., 2003*) and recruitment to the leading edge (*Rossé et al., 2006*); by its direct association with the WRC complex (*Biondini et al., 2016*), exocyst drives WRC to the leading edge, where WRC stimulates actin polymerization, protrusion formation, motility and invasion.

Unexpectedly, using the model of isogenic cell lines HEK-HT and HEK-HT-H-RasV12, we found that hyper-activations of MAP kinase and PI3 kinase pathways were not required for Ras-driven invasion. A huge previous literature had undoubtedly shown that MAPK and PI3K are involved in migration and invasion of various normal and cancer cell models (*Keely et al., 1997*; *Klemke et al., 1997*; *Ward et al., 2001*; *Janda et al., 2002*; *Campbell et al., 2007*). However we speculate that this could not be the case in specific, relevant cellular contexts, such as the HEK-HT-H-RasV12 cells in which the RalB pathway appears to be the dominant driver of invasion down-stream oncogenic Ras. In support of this hypothesis, in a panel of pancreatic cell lines, Ral pathway was found to be more

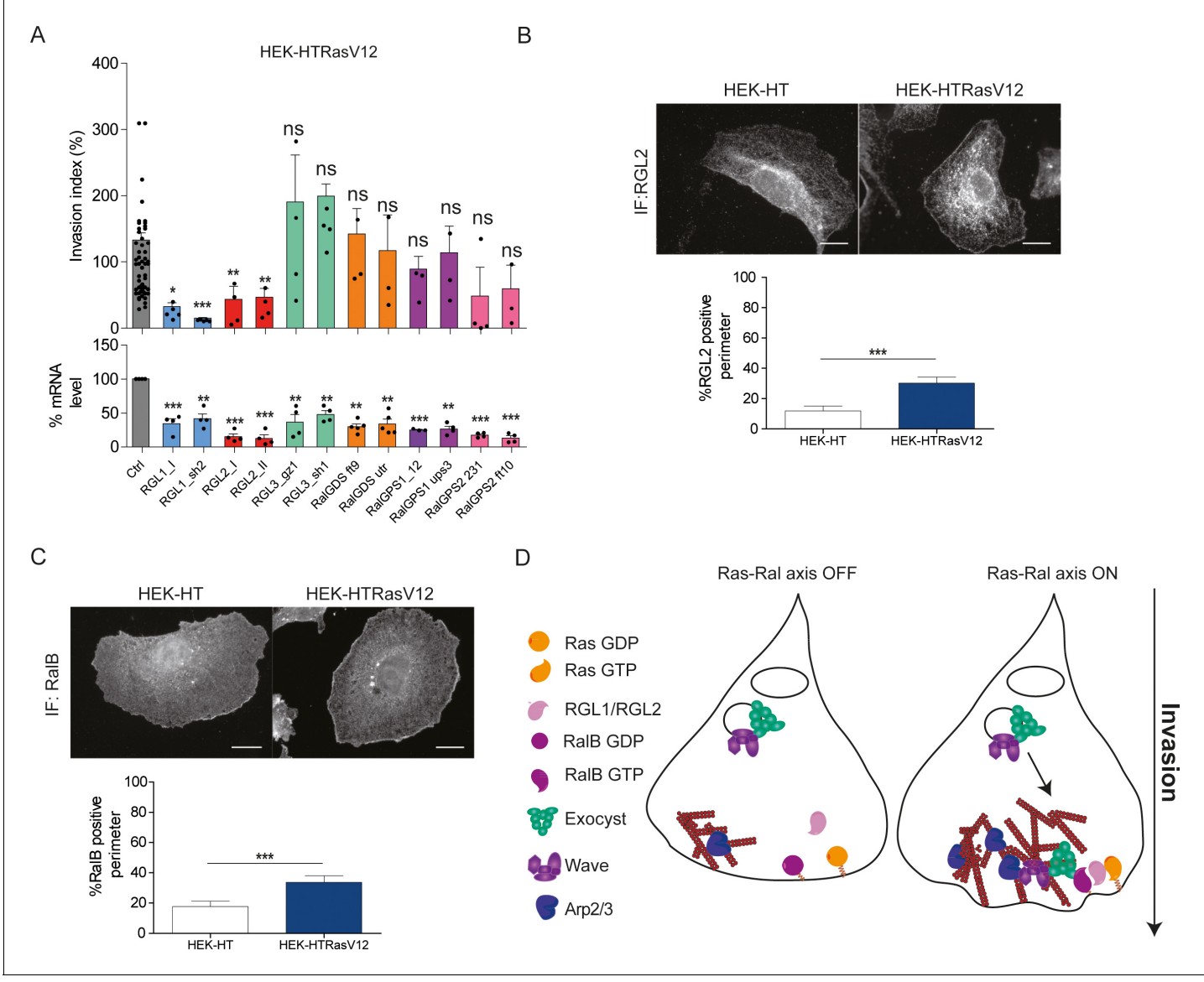

**Figure 6.** Ras-driven invasion requires the two RalGEF RGL1 and RGL2. (A) HEK-HTRasV12 cells require RGL1 and RGL2 for Transwell invasion (upper panel). Depletions of the six RalGEF were validated by RT-qPCR (lower panel). Graphs show the mean ±SEM. Each dot represents one experiment. For statistics one-way ANOVA was used. *p<0.05, **p<0.01, ***p<0.001, ns not-significant. (B–C) Localization of endogenous RGL2 and RalB is increased at the cell edges of HEK-HTRasV12 transformed cells as compared to HEK-HT. Representative epifluorescence images of the immunostaining of RGL2 and RalB are depicted. Scale bar, 10 µm. Graphs show the averaged percentage ±SEM of cell perimeter which is positive for RGL2 or RalB. n > 40 cells from four independent experiments. For statistics Student t-test was used. *p<0.05, **p<0.01, ***, p<0.001, ns indicates not-significant. (D) The Ras-RGL1/2-RalB-exocyst-WRC axis drives invasion. Active Ras-GTP binds and recruit the two RalGEFs RGL1 and RGL2, which activate RalB; RalB-GTP binds to the exocyst complex, promoting its assembly and recruitment to the leading edge; by its direct association with the WRC complex, exocyst drives WRC to the leading edge, where WRC stimulates actin polymerization via the Arp2/3 complex, consequently promoting protrusion formation, motility and invasion.

DOI: https://doi.org/10.7554/eLife.40474.020

commonly activated as compared to MAPK and PI3K, since abnormally high Ral-GTP levels were

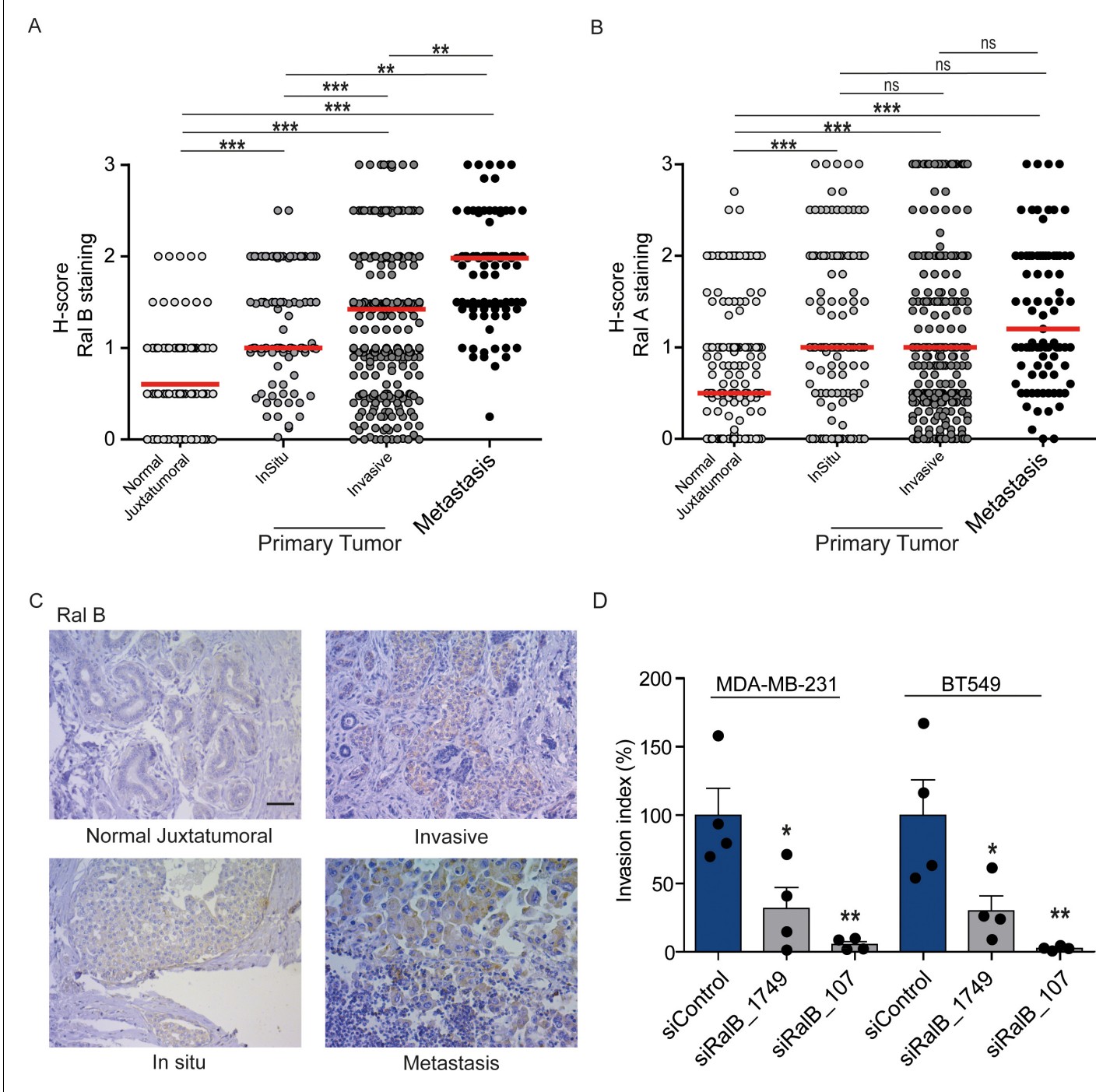

**Figure 7.** Expression of Ral proteins in human breast cancers. Role of RalB in breast cancer invasion. (**A**) Quantitative analysis of RalB protein level in patient samples of normal juxtatumural tissue (n = 298), in situ carcinoma (n = 101), invasive ductal carcinoma (n = 439) and lymph node metastasis (n = 91). Since none of the data sets passed the Shapiro-Wilk normality test, the median H-scores are displayed. Wilcoxon signed-rank tests were performed on paired (i.e. from same patient) data points: n = 73 for normal versus in situ, n = 290 for normal versus invasive, n = 63 for normal versus metastasis, n = 99 for in situ versus invasive, n = 29 for in situ versus metastasis, n = 85 for invasive versus metastasis. *p<0.05, **p<0.01, ***p<0.001, ns not-significant. (**B**) Quantitative analysis of RalA protein level in patient samples of normal juxtatumural tissue (n = 371), in situ carcinoma (n = 144), invasive ductal carcinoma (n = 462) and lymph node metastasis (n = 87). Since none of the data sets passed the Shapiro-Wilk normality test, the median H-scores are displayed. Wilcoxon signed-rank tests were performed on paired (i.e. from same patient) data points: n = 129 for normal versus in situ, n = 368 for normal versus invasive, n = 69 for normal versus metastasis, n = 142 for in situ versus invasive, n = 39 for in situ versus metastasis, n = 83 for invasive versus metastasis. *p<0.05, **p<0.01, ***p<0.001, ns not-significant. (**C**) Representative RalB immunostaining in the different breast cancer compartments. Scale bar, 250 μm. (**D**) Invasion of human breast cancer cells requires RalB, regardless of Ras mutational status. MDA-MB-231 cells

*Figure 7 continued on next page*

*Figure 7 continued*

(carrying the KRasG13D mutation) and BT549 cells (Ras wild-type) were transfected with the indicated siRNAs and subjected to Transwell invasion assay. Graph shows the mean ±SEM. Each dot represents one experiment. For statistics one-way ANOVA test was used. *p<0.05, **p<0.01, ***p<0.001. ns not-significant. Legends for figure supplements.

DOI: https://doi.org/10.7554/eLife.40474.021

The following figure supplement is available for figure 7:

**Figure supplement 1.** Ral proteins and breast cancers.

DOI: https://doi.org/10.7554/eLife.40474.022

much more frequent than abnormally high levels of phosphorylated ERK1/2 or phosphorylated AKT (*Lim et al., 2005*).

In human cancer contexts, the contribution of Ral pathway activation is likely still under-estimated. Even though genetic alterations in Ral genes are rare events in human cancers, occurring in only 1.5% to 2% of patient cases considering a variety of cancer types, alterations in genes coding for Ral regulators in human tumors are not uncommon. Notably, RGL1 and RalGPS1 are found amplified in 11% of breast cancer patients, meanwhile RalGAPA1, RalGAPA2 and RalGAPB are altered in 14%, 8% and 6% of lung cancer patients, respectively (cbioportal bioinformatics platform) (*Gao et al., 2013*), consistent with the notion that dysregulation of Ral might be important for oncogenesis or tumor progression.

In almost all cancer types examined (pancreas, colon, lung, bladder, prostate, melanoma), increased overexpression and/or activation of both RalA and RalB have been observed in patient tumor samples compared with normal tissues, regardless of their Ras mutation status (*Yan and Theodorescu, 2018*). Moreover, Ral-GTP level were found elevated in various tumor-derived cell lines harboring different Ras status, including pancreas (*Lim et al., 2005*), colon (*Martin et al., 2011*), bladder (*Saito et al., 2013*), liver (*Ezzeldin et al., 2014*), lung (*Male et al., 2012*) and brain (*Ginn et al., 2016*). The only exception so far is squamous cell carcinoma (SCC), where RalA was found to suppress rather than promote tumor progression (*Sowalsky et al., 2010*).

In this work we examined Ral protein expression in patient breast tumor samples and we found overexpression of both RalA and RalB as compared to normal breast tissues, therefore adding breast cancers to the list of human cancers with potential overstimulation of Ral pathway. Very interestingly, RalB (but not RalA) protein expression increased in a manner consistent with disease progression (normal < in situ < invasive < metastatic tissues), supporting once more a crucial role for RalB in cancer invasion and metastasis.

In conclusion, we propose that the pharmacological inhibition of the here established Ras-RGL1/2-RalB-exocyst-WRC pathway holds promises as anticancer strategy and definitely warrants further investigations.

## Materials and methods

### Cell culture and transfection

Cells were grown in Dulbecco's modified Eagle's medium (DMEM) supplemented with 2 mM glutamine, penicillin, streptomycin, 10% fetal bovine serum. All cell lines were systematically tested to exclude mycoplasma contamination using a qPCR-based method (VenorGem Classic, BioValley) and authenticated by SRT profiling (GenePrint 10 system, Promega). See *Supplementary file 1* for detailed information and selection antibiotics. Transient DNA transfections were performed with Lipofectamine Plus reagents (Invitrogen) or JetPRIME (Polyplus). Transient double siRNA and DNA transfections were performed with RNAiMAX (Invitrogen) and JetPEI (Polyplus). See *Supplementary file 2* and *Supplementary file 3* for plasmid and siRNA sequences used in this work.

### Lentiviral transduction and cell sorting

All cells lines stably expressing the optogenetic dimerizing system were generated from HEK-HT cell lines via infection using lentiviruses. For viral production, 293 T cells were transfected with Lipofectamine 2000 (Invitrogen) with pHR-CRY2 or pLVX viral vectors along with lentiviral packaging plasmid

(psPAX2) and VSVG expression vector (pCMV-VSV-G). Viral supernatant was harvested at 72 hr post-transfection, filtered, and added to the recipient cell lines for 24 hr with 6 µg/ml polybrene (# 107689, Sigma-Aldrich). Cells were sorted for co-expression of the fluorescent constructs by cell sorting on a BD FacsAria II (BD Bioscience, Flow Cytometry Core Facility Institut Curie), and cell populations were used in bulk.

## Live cell imaging

For imaging cells were plated on 25 mm glass coverslips, pre-coated with 120 µg/mL of collagen type I of rat-tail (Institut de Biotechnologie Jacques Boy), and placed in an observation chamber (Pecon, Meyer Instruments). Imaging was performed at 37°C and 5% CO2 in a heating chamber (Pecon, Meyer Instruments) on an inverted microscope Olympus IX71 equipped with a 100X objective with NA 1.45. The microscope was controlled with the software Metamorph (Molecular Devices). Differential interference contrast (DIC) imaging was performed with a far-red filter (cutoff 550 nm, BLP01-635R-25, Semrock) in the illumination path to avoid CRY2 activation. Total internal reflection fluorescence (TIRF) images were acquired using an azimuthal TIRF module (ilas2, Roper Scientific). Localized stimulation was performed with the FRAP system at low laser power (~5%) with both 491 nm and 401 nm wavelength light lasers (Stradus), 2–3 100 ms pulses, every 10–15 s. Imaging of Cherry and iRFP was done with 561 nm and 642 nm laser (Stradus), respectively. For image processing and data analysis, after background subtraction, the mean fluorescence over the time inside the blue light activated region (ROI) was measured using Metamorph software. The mCherry and iRFP recruitment curves obtained from TIRF images were expressed as fold increase of fluorescence: the value of fluorescence for each time point was normalized by the pre-illumination fluorescence (mean of the first frames without light activation). For cell edge morphodynamics and recruitment analysis, a custom-built ImageJ plugin, named 'Recruitment Edge Dynamics', was used (Paul-Gilloteaux et al., 2018).

## Transwell invasion assay

The Transwell insert consisted in a porous membrane (8 µm pore size) with on top a thin layer of matrigel matrix that mimics the extracellular matrix (#354483, BD Biosciences). Prior to seeding in the Transwell chambers, cells were starved overnight in DMEM with 0.2% serum. 100,000 cells per well were seeded on the top of the insert in 0.2% serum medium, while 10% serum medium was placed in the well below, to create a chemoattractant gradient. After 6 hr of incubation at 37°C and 5% $CO_2$, the insert was washed with PBS, non-invasive cells on the top of the insert were removed with a cotton swab. The cells that did invade on the bottom were fixed with 4% PFA, permeabilized with 0.5% NP-40 and incubated with 5 mg/ml DAPI to stain the nuclei. The porous membranes were cut out and mounted on slides using ProLong Gold antifade reagent (Invitrogen). Cell nuclei were imaged using an epifluorescence a Zeiss Axioplan microscope (10X objective), images were acquired with a Coolsnap HQ2 camera (Roper Scientific) and counted using ImageJ plugin 'Cell Counter'. The Invasion index is calculated from the mean of all the replicates per each condition, normalized on the mean of the control per each experiment. For Transwell coupled with optogenetic, each well was illuminated with an independent LED using a custom-made illumination device consisting of an array of 5 mm blue LEDs (480 nm, 12 cd, 30°) driven by an Arduino Due micro-controller and delivering 3.5 mW each. Blue light was administered continuously from the bottom of the Transwell plate for the whole duration of the experiment (6 hr).

## Inverted invasion assay

Inverse invasion assays were performed accordingly to a previously published protocol (Kajiho et al., 2016), but using a collagen gel instead of Matrigel. An ice-cold gel preparation of bovine collagen type I (#5005-B, Advanced Biomatrix) at a final concentration of 2.3 mg/mL, in MEM medium containing 0.28% NaHCO3 (pH = 8), was incubated 2.5 hr at 4°C (nucleation step), pipetted into 12-well 8-µm-pore-diameter transwells (#353182, Corning) (230 µm gel preparation per insert), inserted into a 12-well tissue culture plate, and incubated at 37°C for 1 hr (polymerization step). The transwells were then inverted, $4 \times 10^4$ cells were seeded on the underside of the porous membrane, and placed in 6-well plate for 2 hr (cell attachment step). The transwells were inverted back, washed 3 times with 1 ml of serum-free medium, and placed in 1.5 ml of serum-free medium inside a 12-well

plate (lower chamber); 230 µl of 10% FBS-DMEM supplemented with 25 ng/ml EGF were added inside the transwell (upper chamber). The cells were then allowed to invade upward into the bovine collagen and toward the gradient of serum/EGF for 4 days at 37°C and 5% $CO_2$. Cells were live stained with CellTrace yellow (Thermofisher) for 1 hr at 37°C and rapidly imaged by confocal microscopy (Zeiss LSM880, 10x objective, excitation at 545 nm and emission at 605 nm). Optical sections were captured at 5 µm intervals, starting from the underside of the transwell membrane and moving upward in the direction of cell invasion. The resulting fluorescence images were quantified using ImageJ software. For each independent experiment, data were generated from two duplicate transwells, and optical sections were acquired from two areas of each transwell.

## Immunoblotting, Immunofluorescence, RT-qPCR

For immunoblotting, cells were lysed in RIPA buffer (150 mM NaCl, 2 mM MgCl, 2 mM CaCl2, 0.5% NaDOC, 1% NP40, 0.1% SDS, 10% Glycerol, 50 mM Tris-HCL pH 8.0) containing 2 mM Na3VO4, 10 mM NaF, 1 mM DTT and a protease inhibitor mixture (#0589291001, Roche). Equal amounts of protein were diluted in 4x Laemmli buffer and resolved by SDS-PAGE. Proteins were transferred to 0.45 µm nitrocellulose membranes (Whatman) by wet transfer and blocked with 3% BSA in TBS/0.05% Tween-20 for 30 min. Primary antibodies were: mouse anti-RalA (#610222, BD Transduction Laboratories, dilution 1:1000); rabbit anti-RalB (#3523, Cell Signaling, dilution 1:500); mouse anti AKT(pan) (#2920, Cell Signaling, dilution 1:1000); rabbit anti p-AKT(Ser437)(#9271, Cell Signaling, dilution 1:1000). Protein levels were detected using LICOR Odyssey Infrared Imaging System (LI-COR Biosciences) upon incubation with IRDye secondary antibodies for 1 hr at room temperature.

For immunofluorescence, cells were cultured on coverslips, fixed with 4% paraformaldehyde, quenched with 1M Glycine solution, permeabilized with 0.1% of Triton-100X, incubated with 4% FBS +1% BSA blocking solution, then with primary and secondary antibodies, every step being in PBS buffer. Primary antibodies were: mouse anti-RalB (#WH0005899M4, clone 4D1, Sigma-Aldrich, dilution 1:200), mouse anti-RGL2 (#H00005863-M02, clone 4D10, Novus biologicals, dilution 1:200). For RT-qPCR analysis, total RNA was extracted using RNaeasy Plus Mini kit (Quigen). Retrotranscription and amplification were obtained using the iScript cDNA synthesis kit (BioRad), the SYBR Green Master Mix kit or TaqMan (Applied Biosystems) on the ABI Prism7900 SequenceDetection System (Perkin-Elmer Applied Biosystems). See *Supplementary file 4* for the RT-qPCR primers.

## Rac1 and RalB activity measurements

Rac1 activity was measured using the 'Rac1 Pull-down Activation Assay Biochem Kit' (# BK035, Cytoskeleton), which is based on beads-coupled GST-Pak1GBD. A similar pull-down assay was used for RalB activity but using home-made beads-coupled GST-Sec5GBD. For RalB activity measurement by FRET, cells were transfected with a validated FRET-based RalB biosensor (*Martin et al., 2014*) and imaged with an Olympus IX71 microscope equipped with a FRET filter set (86002v1JP4, Chroma Technology) and a 63x oil-immersion objective. YFP/CFP images were generated using Metamorph and ImageJ software.

## Size exclusion chromatography

Cells were washed in ice-cold PBS and lysed in 30 mM Tris (pH 7.2), 150 mM NaCl and 1% CHAPS. Lysates were clarified by centrifugation at 40 000 r.p.m. for 30 min. Protein was fractionated on a Superose 6 10/300 GL Increase (GE Healthcare Life Sciences) connected to an ÄKTA pure chromatography system (GE Healthcare Life Sciences, Protein Expression and Purification Core Facility, Institut Curie). The column was eluted with 30 mM Tris (pH 7.2), 150 mM NaCl and 1% CHAPS at 0.5 ml/min and 0.5 ml fractions were collected. Prior to SDS-PAGE analysis, proteins were concentrated with 20% Trichloroacetic Acid Protein (TCA), washed in Acetone and resuspended in Laemmli buffer.

## Breast cancer samples

Primary tumors and lymph nodes were surgically removed before any radiation, hormonal or chemotherapy, using a cohort of patients (n = 649) treated at the Institut Curie from 2005 to 2006. In this cohort, 502 invasive ductal carcinomas were selected for Ral detection; after discarding the samples with staining problems, the final numbers of analyzed cases were n = 466 for RalA and n = 448 for RalB. Tissue microarrays (TMA) consisted of replicated tumor cores (1 mm diameter) selected from

whole-tumor tissue sections and a matched tissue core from adjacent non-tumoral breast epithelium (referred to as normal breast tissue). Immunoistochemistry (IHC) staining was performed using a Dako Autostainer Plus according to previously published protocols (*Lodillinsky et al., 2016*). Antibodies were: anti-RalB (#WH0005899M4, clone 4D1, Sigma-Aldrich), anti-RalA (#610222, BD Transduction Laboratories). H-scores were calculated using the following formula: stained cell percentage x stain intensity/100.

Analysis of human samples was performed in accordance with the French Bioethics Law 2004–800, the French National Institute of Cancer (INCa) Ethics Charter, and after approval by the Institut Curie review board and ethics committee (Comité de Pilotage du Groupe Sein) that waived the need for written informed consent from the participants. Women were informed of the research use of their tissues. Data were analyzed anonymously.

## Statistics

Results are shown as mean ± standard deviation (SD) or standard error of the mean (SEM). Statistical analysis was performed using Graphpad Prism (v5.0) and R Software v.3.3.2 (*R Core Team, 2016*). Comparisons between two groups were assessed using Student t-test. Comparisons between more than two groups were assessed using one-way ANOVA test. Comparisons between paired data were assessed using Wilcoxon signed-rank test. For breast cancer H-score analysis, Shapiro-Wilk normality test and non-parametric Mann-Whitney tests and Wilcoxon signed-rank test for paired data were used. p values less than 0.05 were considered significant.

## Acknowledgements

We thank the staff of Flow Cytometry Core Facility of Institut Curie for their excellent assistance, the Cell and Tissue Imaging (PICT-IBiSA), Institut Curie, member of the French National Research Infrastructure France-BioImaging (ANR10-INBS-04), Chiara Vicario for discussion and analysis help, and Hiroaki Kajiho and Giorgio Scita for very kindly sharing their detailed protocols. This work was supported by Fondation ARC pour la Recherche sur le Cancer (PJA 20151203371 to MCP, fellowship to GZ), Institut national de la Santé et de la Recherche médicale (INSERM ITMO Plan Cancer 2014–2018, PC201530 to MC and MCP), Association Christelle Bouillot, French National Research Agency (ANR) Paris-Science-Lettres Program (ANR-10-IDEX-0001–02 PSL).

## Additional information

### Funding

| Funder | Grant reference number | Author |
| --- | --- | --- |
| Fondation ARC pour la Recherche sur le Cancer | PJA 20151203371 | Maria Carla Parrini |
| Institut National de la Santé et de la Recherche Médicale | PC201530 | Mathieu Coppey |
| Agence Nationale de la Recherche | ANR-10-IDEX-0001-02 PSL | Mathieu Coppey |

The funders had no role in study design, data collection and interpretation, or the decision to submit the work for publication.

### Author contributions

Giulia Zago, Conceptualization, Formal analysis, Investigation, Visualization, Methodology, Writing—original draft, Writing—review and editing; Irina Veith, Formal analysis, Validation, Investigation, Writing—review and editing; Manish Kumar Singh, Saori Takaoka, Formal analysis, Investigation, Writing—review and editing; Laetitia Fuhrmann, Resources, Data curation, Formal analysis, Writing—review and editing; Simon De Beco, Resources, Methodology, Writing—review and editing; Amanda Remorino, Methodology, Writing—review and editing; Marjorie Palmeri, Frédérique Berger, Formal analysis, Writing—review and editing; Nathalie Brandon, Validation, Investigation; Ahmed El Marjou, Investigation, Methodology, Writing—review and editing; Anne Vincent-Salomon, Resources,

Funding acquisition, Project administration; Jacques Camonis, Conceptualization, Writing—review and editing; Mathieu Coppey, Conceptualization, Supervision, Funding acquisition, Writing—review and editing; Maria Carla Parrini, Conceptualization, Supervision, Funding acquisition, Writing—original draft, Project administration, Writing—review and editing

### Author ORCIDs
Giulia Zago (ID) http://orcid.org/0000-0001-8280-1589
Maria Carla Parrini (ID) http://orcid.org/0000-0002-7082-9792

### Decision letter and Author response
Decision letter https://doi.org/10.7554/eLife.40474.030
Author response https://doi.org/10.7554/eLife.40474.031

## Additional files
### Supplementary files
• Supplementary file 1. List of cell lines
DOI: https://doi.org/10.7554/eLife.40474.023
• Supplementary file 2. List of plasmids
DOI: https://doi.org/10.7554/eLife.40474.024
• Supplementary file 3.: List of siRNAs
DOI: https://doi.org/10.7554/eLife.40474.025
• Supplementary file 4. List of primers
DOI: https://doi.org/10.7554/eLife.40474.026
• Transparent reporting form
DOI: https://doi.org/10.7554/eLife.40474.027

### Data availability
All data generated or analysed during this study are included in the manuscript and supporting files.

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
