## [Decision Letter]

[Editors’ note: a previous version of this study was rejected after peer review, but the authors submitted for reconsideration. The first decision letter after peer review is shown below.]

Thank you for submitting your work entitled "RalB directly triggers invasion downstream oncogenic Ras by mobilizing the Wave complex" for consideration by *eLife*. Your article has been reviewed by three peer reviewers, and the evaluation has been overseen by a Senior Editor/Reviewing Editor. The reviewers have opted to remain anonymous.

Our decision has been reached after consultation between the reviewers. Based on these discussions and the individual reviews below, we regret to inform you that your work will not be considered further for publication in *eLife*.

As you will see, all three reviewers recognise the significant potential interest of the work. At the same time, they also identify several limitations of the study. Very concisely, as the detailed reviews are pasted below, the major concerns fall into the following categories:

1) Concerns that Transwell invasion assay may not be a fair reporter of invasive behaviour of cancer cells.

2) Concerns that observed effects may reflect possible mislocalization of the probes.

3) Concerns that sample sizes might be too small to support or deny effects.

4) Concerns whether the observed effects apply to cancer cells or to normal/physiological context.

5) Concerns of over-reliance on a single approach without further validation with orthogonal approaches.

We are afraid that addressing these concerns thoroughly would take in excess of approximately two months, the time limit for revision at *eLife*, and therefore we prefer to return the manuscript to you so that you can decide how to proceed with the manuscript farther.

*Reviewer #1:*

Zago et al. investigate the mechanisms whereby oncogenic Ras drives invasion. They provide data pointing to RalB as the main driver of invasiveness via the RalGEFs RGL1 and RGL2, independently of MAPK and PI3K. For this, they utilize a novel optogenetic approach, whereby light-controlled local activation of Ral promoted, via its effector exocyst, the recruitment of the Wave Regulatory Complex at the cell leading edge in a Rac1-independent fashion. There, the WRC stimulates protrusion formation. Authors show that illumination-driven Ral activation is sufficient to trigger cell invasion.

My main criticism for this study is that most of it (if not all) relies in the fanciful, though artificial, optogenetic system to activate Ral. In the complete absence of some alternative approach to verify its results.

This optogenetic system should simulate Ras-induced activation of Ral. However, in this particular case, the approach taken is rather inconsistent: They use H-RasV12 to transform HEK-HT cells. There, by using Ras effector mutants, they identify the RalGEF pathway as the inductor of invasion. And they utilize this same cellular setting to identify the GEFs RGL1 and RGL2 as the culprits, using a siRNA screen. Then they use the optogenetic system, based on the GEF domain of RGL2, for all subsequent experiments. It is on the Ras > optogenetic system connection where the main shortcomings are found:

1) For the initial observation they use H-RasV12. However, they use KRas CAAX box to target the optogenetic system to the plasma-membrane. Why do they assume that H-Ras proinvasive signal using RGL1/2 comes from the plasma-membrane? It is well known that H-Ras is found in many membrane systems; why not from endomembranes or endosomes? It could very well be that their optogenetic system, targeted to endomembranes or endosomes evokes a completely different biological outcome. Some attempt to study RGL1/2 sublocalization in H-RasV12-transformed cells would help in this matter.

2) In this respect, the extent to which the optogenetic system using RGL2 Cdc25 domain resembles "total" RGL2 is doubtful. The Cdc25 domain deprived of its regulatory sequences loses all its "personality". This reviewer would bet that the same optogenetic system using another Cdc25 domain (even that one of RalGPS1 that does not bind Ras) would yield a very similar outcome (this control must be done). Thus, ultimately, what authors are observing is invasiveness driven by Ral from some insufficiently characterized sublocalization.

3) Following this line of argument, it would help their case very much if they showed whether overexpression, or inducible expression, of RGL1/2 mimicked the results obtained with the optogenetic system with respect to the exocyst > WRC recruitment.

*Reviewer #2:*

The manuscript by Zago et al. describes an unexpected central role of RalB in Ras-driven invasion and supports earlier findings about a specific role of RalB, and not of RalA, in cell motility. By using an innovative optogenetic approach they finely dissect the interaction cascade (RalGEF-Ral-Exocyst-WRC) with unprecedented spatial-temporal resolution. The manuscript is generally well written and the experiments appear robustly controlled.

As a general remark, the authors should discriminate better to which extent their findings apply to a normal physiologic or cancer context, and in the last case, whether they are true for Ras mutant tumor cells, Ras WT tumor cells or both. Unfortunately the fact that part of the experiments are performed in a Ras mutant context and part in a Ras WT context, and that no validation experiments are performed in human cancer cell lines, makes this distinction difficult. I encourage the authors to either validate the role of RalB in invasion in few (Ras mutant and Ras WT) human cancer cell lines (for example by CRISPR knockout of RalB, followed by a Transwell invasion assay), or to mitigate their speculations about the link between RalB and migration in a Ras mutant and tumor context.

In Figure 1C, the authors show that silencing of RalB, but not RalA, impaired invasion of HEK-HTRasV12 cells of approximately 60%. The authors do not comment on the remaining 40% invasiveness that is not inhibited by RalB suppression. It would be interesting to test whether combined RalA and RalB silencing is able to completely suppress invasion, suggesting that although RalB is the main mediator of invasiveness, RalA is able to take over in the absence of RalB.

In Figure 3D, the authors do not mention how long the cells were treated with the Rac inhibitor NSC23766. In addition to provide this information, the authors should prove that Rac1 was inhibited in their cells at the dose and time used in their experiment.

In Figure 5A, the authors show that RalB activation is sufficient to promote invasion by recruiting the WRC via the exocist. At the end of their result description the authors conclude that "since the OptoRal cells do not carry Ras mutations, this experiment shows that RalB activation is able to drive cell invasion even in a wild-type Ras genetic context". The authors can't make this conclusion, because in their system Ral activation is artificially made independent of Ras activation and mutation status (unless they can quantitatively compare the level of Ral activation following illumination in their OptoRal cells with the levels of endogenous Ral activation in the presence of WT or mutant Ras).

In Figure 5A, the authors should include OptoRal and OptoControl cells (without iRFP-Abi1) to show if invasion is increased after illumination-induced Ral activation in the presence of endogenous levels of Abi1, or if overexpression is needed.

*Reviewer #3:*

In this manuscript Zago et al. report that Ral activation causes recruitment of the Scar/WAVE complex via the exocyst complex. The authors describe the development of an elegant novel optogenetics approach to recruit a RalGEF to the plasma membrane upon blue light illumination thereby activating Ral proteins at this location. They report that this causes recruitment of the Scar/WAVE complex and causes consequently membrane protrusion independent of Rac activity. They further claim that active Ras induced cell invasion depends on Ral but not MAPK or PI3K activity and is mediated by RalGEFs RGL1/2. Finally, they suggest that Ral mediated increase in cell invasion is mediated by the exocyst recruiting the Scar/WAVE complex.

This is potentially a very exciting story: Even though many regulatory mechanisms of the Scar/WAVE complex are known, it is still not clear how its activity is controlled in time and space during lamellipodia protrusion and cell migration and invasion. In general, the study is done with great care and well controlled. Therefore, I would be willing to support the publication of this manuscript in *eLife* if my major concerns are adequately addressed:

1) Transwell invasion assays used in Figure 1 and 5 have inherent shortcomings: It is an endpoint analysis which does not allow visualization of the invasion. Thus, it does not control for a defect or increase in adhesion and these could be interpreted as a decrease or increase in invasion. Inverted invasion assays are more reliably controlled in vitro invasion assays.

2) In Figure 1B: It is quite a big statement that Ras driven invasion is mediated by Ral from this one Transwell assay. This needs to be backed up by data from other assays for example, inverted invasion assays using knockdown, MAPK pathway inhibitors etc. The spread of the mean between individual experiments is huge and does not give confidence in the reliability of the Transwell experiment.

3) Figure 4B, D: Why does the iRFP-Abi in the OptoRal cells does not localise to the leading edge of cells as one would expect if it is incorporated into the Scar/WAVE complex? This is seen for the iRFP-Abi in the OptoControl cells. An increase of the iRFP-Abi in the OptoRal cells at the leading edge and not just at the membrane underneath the cell would increase confidence that RalB-exocyst activation contributes to a physiological recruitment of the WRC to the leading edge of cells. Could you activate a smaller area only at the leading edge of cells with blue light? Alternatively, you could use mesenchymal cells that form larger lamellipodia. In addition to Abi another component of the Scar/WAVE besides Abi should be imaged since Abi can interact with N-WASP independently of the Scar/WAVE complex.

4) For some of the experiments more appropriate statistics should be used (see below).

Figure 1B-E: Repeated t-test is inappropriate. The appropriate statistics to compare several groups is One-way ANOVA.

Figure 1C: Please show a western blot.

Supplementary Figure 1A-B: The information for the statistics used is missing.

Figure 1C, E: It is more appropriate to normalise the control siRNA values in a way to retain the SD and not to arbitrarily set them to a fixed value of 100%. This will influence your statistics in an inappropriate way. This can be done in PRISM using the normalise function and setting the 0% to 0 and 100% to the mean of your control.

Figure 1E: RalGPS2 has a big spread between the means whereas RGL1/2 have a small spread. Doing more repeats for RalGPS2 may also reveal a significant reduction. Therefore, it is premature to conclude that the Ras mediated activation of RalB is mediated only by RGL1/2.

Video 2 appears to be the same as Video 1. Is this a mistake?

Figure 2B: Is this increase in protrusion formation dependent on Ral activation? Use CRY2-mCherry as control. This should not lead to protrusion formation. I guess this was done in Figure 3D? Does OptoControl stands for CRY2-mCherry control?

Figure 2E: How reliable is the Sec5GBD-iRFP or PakGBD recruitment as a tool to measure Ral and Rac activity? This should be confirmed using a FRET biosensor or pulldown assays of active Rac or RalB.

Figure 1—figure supplement 1C: What is the extra band at 110 kDa in the mCherry blot that shows up in both Opto control and Ral lysates?

Figure 3: The Recruitment Edge dynamics and MATLAB based segmentation method needs to be described better since it is unpublished unless the cited in press paper is available on a preprint server.

Figure 3D: Two-way ANOVA would be the more appropriate statistics here instead of repeated t-tests since several groups are compared. The vehicle control for the Rac inhibitor is missing.

Figure 5A: Two-way ANOVA would be the more appropriate statistics here instead of repeated t-tests since several groups are compared.

[Editors’ note: what now follows is the decision letter after the authors submitted for further consideration.]

Thank you for resubmitting your work entitled "RalB directly triggers invasion downstream Ras by mobilizing the Wave complex" for further consideration at *eLife*. Your revised article has been favorably evaluated by Andrea Musacchio as the Senior/Reviewing Editor, and two reviewers.

The manuscript has been improved but there are some remaining issues that need to be addressed before acceptance, as outlined below:

*Reviewer #1:*

The authors have re-written the study and provided new data to generate a coherent and convincing case.

A minor experiment would be necessary to round up the whole story: authors speculate that the "light-controlled stimulation of cell protrusions is essentially due to the activation of endogenous RalB, rather than RalA". In order to prove this beyond doubt, and to match this data with that shown in Figure 5C for HEK-RasV12 cells, the OptoRal light-controlled stimulation of cell protrusions should be performed using siRNAs for RalA and B.

*Reviewer #2:*

The new version of the manuscript by Zago et al. maintains the merit to address the poorly explored role of Ral proteins as downstream Ras effectors, in particular in cell invasion. The authors apply the innovative optogenetic approach to finely dissect the molecular cascade that links Ras activation to invasion in their specific cellular model.

The new version of the manuscript is substantially improved, both with respect to the clarity of the results presentation and the consistency of the conclusions drawn by the authors.

The additional experiments helped making the data more robust and clarifying that the RalGEF-Ral-Exocyst-WRC cascade appears relevant in the tumor context although independent of Ras mutational status.

---

## [Author Response]

[Editors’ note: the author responses to the first round of peer review follow.]

We thank the three reviewers for their constructive criticisms that helped to direct our efforts to improve the quality of our paper.

A consistent amount of new data was included for a total of 10 new panels in the main figures, and 9 new panels in the supplementary figures.

In particular, we now show clinical data from breast cancer tissue samples, thanks to a fruitful collaboration with the pathology department of Curie Hospital (and consequently a few extra authors). These new data show a correlation between expression at protein level of RalB (and not RalA) and disease progression. This is the first evidence of an implication of Ral pathway in breast cancers, and complements the available literature on the role of Ral in other human cancers, including lung, colon, pancreas, prostate, melanoma.

In such broader human health perspective, it should become more obvious the added value of optogenetics in Ral signaling field. We could move from correlations to causality, 1) demonstrating the sufficiency of RalB activation to trigger invasion, and 2) identifying the underlying molecular mechanisms.

Reviewer #1:[…] My main criticism for this study is that most of it (if not all) relies in the fanciful, though artificial, optogenetic system to activate Ral. In the complete absence of some alternative approach to verify its results.This optogenetic system should simulate Ras-induced activation of Ral. However, in this particular case, the approach taken is rather inconsistent: They use H-RasV12 to transform HEK-HT cells. There, by using Ras effector mutants, they identify the RalGEF pathway as the inductor of invasion. And they utilize this same cellular setting to identify the GEFs RGL1 and RGL2 as the culprits, using a siRNA screen. Then they use the optogenetic system, based on the GEF domain of RGL2, for all subsequent experiments. It is on the Ras > optogenetic system connection where the main shortcomings are found:

We realize that unfortunately the original presentation structure lacked clarity and produced several misunderstandings. We now changed the order as follows.

First, we better introduce the previous literature supporting a role for RalB role in human cancer invasion (Introduction, second paragraph), and we clearly state that the optogenetic approach is a tool to study the underlying molecular mechanisms (subsection “Optogenetic control for selective activation of Ral proteins”, first paragraph). Of course, optogenetic Ral activation must be performed in a wt Ras context, in which Ral is not abnormally stimulated; this is why we chose the HEK-HT cell model. We engineered our constructs in a way to activate Ral at the plasma-membrane because the literature indicates that Ral oncogenic signaling emanates at least in part from the plasma-membrane: the membrane-targeted murine RalGEF Rlf-CAAX transforms HEK-HT and BJ-HT human cells (Hamad et al., 2002; Lim et al., 2005); the expression of membrane-targeted human RalGDS-CAAX promotes metastasis of murine 3T3 cells in a tail vein injection assay (Ward et al., 2001).

Second, we describe the optogenetic experiments and conclusions on the cause-effect link between Ral activation (at the plasma-membrane), WRC recruitment, protrusion formation, and invasion (now adding results from a second in vitro invasion assay, new Figure 5B, 5D).

Third, we evaluate the relative contributions of Ral, MAPK and PI3K pathways, in an oncogenic Ras context, the HEK-HT-H-RasV12 cell model, finding that RalB pathway appears to be the major driver of Ras-dependent invasiveness, as shown by the V12G37 mutant and by pharmacological inhibitors (new Figure 5F).

Fourth, by a siRNA screen, we identified RGL1 and RGL2 as the missing molecular link between H-RasV12 and RalB for invasion. Consistently, we now show that oncogenic Ras drives recruitment of endogenous RGL2 and RalB at the cell plasma-membrane (new Figure 6B, 6C).

Fifth, we now add clinical data from breast cancer tissue samples, showing that RalB protein expression increases in a manner consistent with cancer progression (normal < in situ < invasive < metastatic tissues) (new Figure 7).

We note that while the application of optogenetics led us to robust mechanistic conclusions, in the revised version a conspicuous amount of the reported results (3 out of 7 figures) do not rely on the optogenetic system.

1) For the initial observation they use H-RasV12. However, they use KRas CAAX box to target the optogenetic system to the plasma-membrane. Why do they assume that H-Ras proinvasive signal using RGL1/2 comes from the plasma-membrane? It is well known that H-Ras is found in many membrane systems; why not from endomembranes or endosomes? It could very well be that their optogenetic system, targeted to endomembranes or endosomes evokes a completely different biological outcome. Some attempt to study RGL1/2 sublocalization in H-RasV12-transformed cells would help in this matter.

We totally agree with the reviewer that optogenetic activation of Ral at endomembranes might lead to different biological outcomes, taking into account for example the crucial role of RalB in autophagy. Indeed, our future efforts go towards this direction. The point of this paper is that activation of Ral at plasma-membrane leads to invasiveness, without excluding other outputs, depending or not on the location of Ral activity.

As proposed by the reviewer, we studied RGL2 (and RalB) subcellular localization in HEK-HT versus H-RasV12 transformed cells: oncogenic Ras drives recruitment of endogenous RGL2 and RalB at the cell plasma-membrane (new Figure 6B, 6C), thus supporting the relevance of our choice to optogenetically activate Ral at the plasma-membrane.

2) In this respect, the extent to which the optogenetic system using RGL2 Cdc25 domain resembles "total" RGL2 is doubtful. The Cdc25 domain deprived of its regulatory sequences loses all its "personality". This reviewer would bet that the same optogenetic system using another Cdc25 domain (even that one of RalGPS1 that does not bind Ras) would yield a very similar outcome (this control must be done). Thus, ultimately, what authors are observing is invasiveness driven by Ral from some insufficiently characterized sublocalization.

There is a big misunderstanding on this point. We never meant to mimic RGL2 signaling using just its catalytic RalGEF Cdc25-like domain. We totally agree with the reviewer that very likely any other catalytic RalGEF Cdc25-like domain would yield a very similar outcome, i.e. protrusion and invasion. The choice of RGL2 was motivated by the fact that its minimal RalGEF Cdc25-like domain has been well characterized biochemically (Ferro et al., 2008), not because it was a hit in our screen. To avoid this misinterpretation we now present the results of the RalGEFs screen after the results of optogenetics. In fact, we exploited a minimal RalGEF domain and removed on purpose the regulatory sequences of RGL2, in order to assess the effects of only Ral activation at the plasma membrane, without stimulation of any other pathway. The fact that we see invasiveness just with a RalGEF domain shows that Ral activation per se (at the plasma-membrane) is sufficient for this outcome.

3) Following this line of argument, it would help their case very much if they showed whether overexpression, or inducible expression, of RGL1/2 mimicked the results obtained with the optogenetic system with respect to the exocyst > WRC recruitment.

We hope that the above explanations clarified the fact that our optogenetic system (at the plasma-membrane, with a minimal RalGEF Cdc25 domain) allows indeed the fine control of when and where we activate endogenous Ral, and for this reason it is superior to an overexpression approach of full-length RGL2 which might lead to uncontrolled/mislocalized Ral hyperactivation and/or dominant negative effects. Any experimental approach is artificial, but optogenetics has the great advantage to discriminate between ‘before’ and ‘after’ the perturbation, in a time scale of a few seconds.

Reviewer #2:The manuscript by Zago et al. describes an unexpected central role of RalB in Ras-driven invasion and supports earlier findings about a specific role of RalB, and not of RalA, in cell motility. By using an innovative optogenetic approach they finely dissect the interaction cascade (RalGEF-Ral-Exocyst-WRC) with unprecedented spatial-temporal resolution. The manuscript is generally well written and the experiments appear robustly controlled.As a general remark, the authors should discriminate better to which extent their findings apply to a normal physiologic or cancer context, and in the last case, whether they are true for Ras mutant tumor cells, Ras WT tumor cells or both. Unfortunately the fact that part of the experiments are performed in a Ras mutant context and part in a Ras WT context, and that no validation experiments are performed in human cancer cell lines, makes this distinction difficult. I encourage the authors to either validate the role of RalB in invasion in few (Ras mutant and Ras WT) human cancer cell lines (for example by CRISPR knockout of RalB, followed by a Transwell invasion assay), or to mitigate their speculations about the link between RalB and migration in a Ras mutant and tumor context.

In the revised version, it is now clear that our findings apply to cancer context. We better introduce the previous literature supporting a role for Ral pathway in various human cancers, with a specific function for RalB in invasion (Introduction, second paragraph). Moreover, we add original data on Ral protein expression in breast cancer tissue samples, supporting for the first time a potential implication of RalB in breast cancer progression (new Figure 7 and Figure 7—figure supplement 1).

The requirement of RalB for Transwell invasion has been previously validated by shRNA knock-down by the group of Chris Counter in seven out of nine K-Ras mutated human pancreatic cancer cell lines (Lim et al., 2006).

We analyzed the effect of RalB silencing by siRNA in two invasive human breast cancer cell lines: MDA-MB-231 (carrying the K-RasG13D mutation) and BT549 (Ras wild-type). RalB was required for invasion in both breast cancer cell models (new Figure 7D), suggesting that RalB might play a role in invasion not only down-stream mutated Ras, but also in Ras wild-type contexts.

In Figure 1C, the authors show that silencing of RalB, but not RalA, impaired invasion of HEK-HTRasV12 cells of approximately 60%. The authors do not comment on the remaining 40% invasiveness that is not inhibited by RalB suppression. It would be interesting to test whether combined RalA and RalB silencing is able to completely suppress invasion, suggesting that although RalB is the main mediator of invasiveness, RalA is able to take over in the absence of RalB.

We performed this experiment and we added the results in Figure 5C. The silencing of both RalB and RalA reached up to 90% invasion inhibition, more than the silencing of RalB alone (60%). This indicates that even though RalA is dispensable for invasion, it might partially compensate when RalB is absent, as correctly anticipated by the reviewer (subsection “Ras-driven cancer invasion requires activation of RalB via RGL1 and RGL2”, second paragraph).

In Figure 3D, the authors do not mention how long the cells were treated with the Rac inhibitor NSC23766. In addition to provide this information, the authors should prove that Rac1 was inhibited in their cells at the dose and time used in their experiment.

The treatment with NSC23766 at 100 mM was started 1 hr before the acquisitions and maintained during the entire experiment; this is now mentioned in the legend of Figure 2D. OptoRal cells, treated at the same dose and time, were subjected to a pull-down assay, showing an inhibition of approximately 50% of the active Rac1-GTP amount (new Figure 2—figure supplement 2C).

In Figure 5A, the authors show that RalB activation is sufficient to promote invasion by recruiting the WRC via the exocist. At the end of their result description the authors conclude that "since the OptoRal cells do not carry Ras mutations, this experiment shows that RalB activation is able to drive cell invasion even in a wild-type Ras genetic context". The authors can't make this conclusion, because in their system Ral activation is artificially made independent of Ras activation and mutation status (unless they can quantitatively compare the level of Ral activation following illumination in their OptoRal cells with the levels of endogenous Ral activation in the presence of WT or mutant Ras).

The sentence was removed. We changed the chapter title to “Activation of Ral-exocyst-WRC axis promote invasion of non-transformed cells”.

In Figure 5A, the authors should include OptoRal and OptoControl cells (without iRFP-Abi1) to show if invasion is increased after illumination-induced Ral activation in the presence of endogenous levels of Abi1, or if overexpression is needed.The experiment was done leading to the expected result (new Figure 4A).Reviewer #3:[…] 1) Transwell invasion assays used in Figure 1 and 5 have inherent shortcomings: It is an endpoint analysis which does not allow visualization of the invasion. Thus, it does not control for a defect or increase in adhesion and these could be interpreted as a decrease or increase in invasion. Inverted invasion assays are more reliably controlled in vitro invasion assays.

Following the recommendation of reviewer #3, we performed several Inverted invasion assays, using 3D collagen I gels (detailed protocol in Materials and methods section), which nicely complemented the standard Transwell assays with a matrigel layer.

We confirmed that the expression of H-RasV12 promotes invasion of HEK-HT cells also in a 3D collagen gel (new Figure 5B), and that RalB is required for this Ras-driven invasiveness (new Figure 5D).

2) In Figure 1B: It is quite a big statement that Ras driven invasion is mediated by Ral from this one Transwell assay. This needs to be backed up by data from other assays for example, inverted invasion assays using knockdown, MAPK pathway inhibitors etc. The spread of the mean between individual experiments is huge and does not give confidence in the reliability of the Transwell experiment.

We believe that the expanded version of Figure 5 (previous Figure 1A, B, C) better support our conclusion that RalB contribution appears to be more important than that of MAPK and PI3K pathways, at least in our genetically controlled cell model.

The inhibitory impact of RalB silencing on HEK-HT-H-RasV12 invasiveness is now shown by Transwell invasion assay (Figure 5C) and by Inverted invasion assay (new Figure 5D).

The previous results with the RasV12G37 mutant (Figure 5E) are not backed up by new data (new Figure 5F) using Trametinib and PIK90 to very efficiently inhibit MAPK and PI3K pathways, respectively. The new results clearly show that HEK-HT-RasV12 cells are still able to invade even when MAPK or PI3K pathways are very efficiently inhibited by the drugs, as assessed by Transwell invasion assay.

We tried a similar drug approach using the Inverted invasion assay. However, since this assay requires much longer time than the Transwell invasion assay (4 days instead of 6 hrs), the strong cytostatic effect of Trametinib and PIK90 on this time scale is a very important bias. Moreover, we observed that the morphology of the cells in the collagen gel was greatly impacted by 4 days treatments. The standard deviations were huge and no significant statistical differences could be identified without versus with treatments (Author response image 1).

3) Figure 4B, D: Why does the iRFP-Abi in the OptoRal cells does not localise to the leading edge of cells as one would expect if it is incorporated into the Scar/WAVE complex? This is seen for the iRFP-Abi in the OptoControl cells. An increase of the iRFP-Abi in the OptoRal cells at the leading edge and not just at the membrane underneath the cell would increase confidence that RalB-exocyst activation contributes to a physiological recruitment of the WRC to the leading edge of cells. Could you activate a smaller area only at the leading edge of cells with blue light? Alternatively, you could use mesenchymal cells that form larger lamellipodia.

The iRFP-Abi does always localize at the protruding leading edges. However, it is sometime tricky to show this sharp localization because of a combination of reasons: i) the leading edge is above the ventral plan which is captured by TIRF; ii) edge movements and Abi1 recruitment present very fast cycling dynamics (protrusion-retraction alternation closely correlating with on-off Abi1 recruitment); iii) iRFP photo-bleaching does occur. Illumination triggers WRC recruitment both at the leading edge and at the ventral side.

We now provide a representative OptoRal cell with a more obvious localization of iRFP-Abi at the leading edge (new Figure 3B and new Video 5). For clarity, we also changed the representative OptoControl (new Figure 3C and new Video 6). In these two cells, iRFP ‘edge’ recruitment was quantified by using an automated method that measures the maximum fluorescent intensity at the edge (width of 1.12 µm) (Paul-Gilloteaux et al., 2018): the resulting heat maps show a dynamic edge fluorescence appearing upon illumination in OptoRal cell (Figure 3B, left) but not in OptoControl cell (Figure 3C, left).

For measurements on several cells and statistical analysis, we chose to quantify iRFP-Abi recruitment at the ‘ventral’ membrane because this was the location where TIRF measurements were precise. HEK-HT cells have pretty large lamellipodia.

In addition to Abi another component of the Scar/WAVE besides Abi should be imaged since Abi can interact with N-WASP independently of the Scar/WAVE complex.

The gel filtration experiments reported in Figure 3—figure supplement 1 indicate that the vast majority (~ 80%) of the exogenous iRFP-Abi co-eluted with the Cyfip subunit in fractions 16-18, corresponding to the expected molecular weight of whole WRC complex (~ 400 kDa). Consequently, the imaged iRFP signal mainly reports the localization of the whole WRC complex.

Moreover, in our experience, it is technically very difficult to use fluorescent fusion with other subunits of WRC complex; for example, GFP-Wave expression seems to produce dominant negative effects.

4) For some of the experiments more appropriate statistics should be used (see below).

As indicated in Statistics paragraph of Materials and methods, we now proceed as follow. “Comparisons between two groups were assessed using Student t-test. Comparisons between more than two groups were assessed using one-way ANOVA test. Comparisons between paired data were assessed using Wilcoxon signed-rank test.”

See below specific answers.

Figure 1B-E: Repeated t-test is inappropriate. The appropriate statistics to compare several groups is One-way ANOVA.

We confirmed the conclusions by applying one-way ANOVA statistical test.

Figure 1C: Please show a western blot.A representative western blot is presented as supplementary Figure 6A.Supplementary Figure 1A-B: The information for the statistics used is missing.

For clarity, since the western blots were highly reproducible, we now show only representative western blots for which we indicate the quantifications below the relevant panels.

Figure 1C, E: It is more appropriate to normalise the control siRNA values in a way to retain the SD and not to arbitrarily set them to a fixed value of 100%. This will influence your statistics in an inappropriate way. This can be done in PRISM using the normalise function and setting the 0% to 0 and 100% to the mean of your control.We thank the reviewer for this useful PRISM tip. We normalized as suggested.Figure 1E: RalGPS2 has a big spread between the means whereas RGL1/2 have a small spread. Doing more repeats for RalGPS2 may also reveal a significant reduction. Therefore, it is premature to conclude that the Ras mediated activation of RalB is mediated only by RGL1/2.

We mitigated our conclusions for RalGPS2: “RalGPS2 silencing showed a potential inhibitory effect, even though statistically not significant”.

Video 2 appears to be the same as Video 1. Is this a mistake?

Yes, it was an uploading mistake for which we apologize. The video was changed.

Figure 2B: Is this increase in protrusion formation dependent on Ral activation? Use CRY2-mCherry as control. This should not lead to protrusion formation. I guess this was done in Figure 3D? Does OptoControl stands for CRY2-mCherry control?

Yes, this was done in Figure 3C, 3D, now Figure 2C, 2D. Yes, OptoControl stands for CRY2-mCherry control. We now make this point clearer by specifying the genetics of OptoRal and OptoControl cells also in panels B and C of Figure 2, in addition to the main text (subsection “Optogenetic control for selective activation of Ral proteins”, first paragraph).

Figure 2E: How reliable is the Sec5GBD-iRFP or PakGBD recruitment as a tool to measure Ral and Rac activity? This should be confirmed using a FRET biosensor or pulldown assays of active Rac or RalB.

Recruitment of GBD domains is a widely used strategy to follow GTPase activities by live imaging. PakGBD-iRFP was used in a previous work (Valon et al., 2015). Sec5GBD-iRFP was cloned for this work, using a Sec5GBD domain (aa 5-97) previously validated for Ral pull-down assays.

We agree that it is important to be sure that our approach activates RalB. We performed additional experiments: both pull-down assays (new Figure 2—figure supplement 1A) and a RalB FRET biosensor indicated (new Figure 2—figure supplement 1B) that light stimulates RalB activity in our OptoRal cells.

Figure 1—figure supplement 1C: What is the extra band at 110 kDa in the mCherry blot that shows up in both Opto control and Ral lysates?

The band is a non-specific band recognized by anti-Cherry antibodies. The band was present also in anti-Cherry western blot from HEK-HT lysates. This is now mentioned in the legend.

Figure 3: The Recruitment Edge dynamics and MATLAB based segmentation method needs to be described better since it is unpublished unless the cited in press paper is available on a preprint server.The paper is now published and the corresponding reference was added: Paul-Gilloteaux et al., 2018.Figure 3D: Two-way ANOVA would be the more appropriate statistics here instead of repeated t-tests since several groups are compared. The vehicle control for the Rac inhibitor is missing.

It is true that in this experiment we have two variables (light/dark and cellular condition). However, for our purposes, we are interested in comparing only the effect of changing light/dark on each cellular condition, so we test only the light/dark variable. In addition, the measurements were paired: same cell pre and post illumination. For these reasons we believe that the most appropriate test is Wilcoxon signed-rank test for paired data.

Moreover, following the suggestion of reviewer #2, we now also compare the difference in the delta velocity pre and post illumination in the presence or absence of the Rac inhibitor and we report in the legend that there is no difference (Student t-test, since we are comparing two groups: deltas in presence and deltas in absence of the Rac inhibitor). The Rac inhibitor was dissolved in water.Figure 5A: Two-way ANOVA would be the more appropriate statistics here instead of repeated t-tests since several groups are compared.

It is true that in this experiment we have two variables (light/dark and cellular condition). However, for our purposes, we are interested in comparing each variable independently. We want to test i) the effect of changing light/dark variable on each cellular condition, ii) the effect of changing cellular condition variable in dark, and iii) the effect of changing cellular condition variable in light. We never compare two groups in which both conditions are changing, for example we do not test OptoControl dark versus OptoRal light. For these reasons we believe that the most appropriate test is One-way ANOVA.

[Editors' note: the author responses to the re-review follow.]

The manuscript has been improved but there are some remaining issues that need to be addressed before acceptance, as outlined below:Reviewer #1:The authors have re-written the study and provided new data to generate a coherent and convincing case.A minor experiment would be necessary to round up the whole story: authors speculate that the "light-controlled stimulation of cell protrusions is essentially due to the activation of endogenous RalB, rather than RalA". In order to prove this beyond doubt, and to match this data with that shown in Figure 5C for HEK-RasV12 cells, the OptoRal light-controlled stimulation of cell protrusions should be performed using siRNAs for RalA and B.

As recommended by reviewer #1, we performed OptoRal light-controlled stimulation of cell protrusions in cells transfected with siRNAs targeting RalA or RalB.

RalB-depleted cells, but not RalA depleted cells, were impaired in protrusions formation, proving that light-controlled stimulation of protrusions is essentially due to activation of endogenous RalB, rather than RalA (new of Figure 2E, new Figure 2—figure supplement 3D).